# Beyond Pixels: Efficient Dataset Distillation via Sparse Gaussian Representation

## Abstract

Dataset distillation has emerged as a promising paradigm that synthesizes compact, informative datasets capable of retaining the knowledge of large-scale counterparts, thereby addressing the substantial computational and storage burdens of modern model training. Conventional approaches typically rely on dense pixel-level representations, which introduce redundancy and are difficult to scale up. In this work, we propose GSDD, a novel and efficient sparse representation for dataset distillation based on 2D Gaussians. Instead of representing all pixels equally, GSDD encodes critical discriminative information in a distilled image using only a small number of Gaussian primitives. This sparse representation could improve dataset diversity under the same storage budget, enhancing coverage of difficult samples and boosting distillation performance. To ensure both efficiency and scalability, we adapt CUDA-based splatting operators for parallel inference and training, enabling high-quality rendering with minimal computational and memory overhead. Our method is simple yet effective, broadly applicable to different distillation pipelines, and highly scalable. Experiments show that GSDD achieves state-of-the-art performance on CIFAR-10, CIFAR-100, and ImageNet subsets, while remaining highly efficient encoding and decoding cost.

## 1 Introduction

Guided by the principles of scaling laws Kaplan et al. (2020), deep learning has advanced along the trajectory of larger models, larger datasets, and longer training, which has significantly pushed the performance boundaries of modern models. However, this paradigm also incurs enormous demands on computation, storage, and communication resources, creating a barrier that hinders both broad adoption and sustainable development. Consequently, a pivotal question in the field is how to achieve strong performance under limited computational and data resources.

Against this backdrop, dataset distillation offers a promising solution. Its goal is to distill the essential knowledge of massive datasets into a compact set of synthetic samples, enabling models trained on these small sets to approximate the performance of those trained on the full dataset. By adopting dataset distillation, the costs of model training, data storage, and data transmission can be substantially reduced. Moreover, it opens up a wide range of applications, including efficient data replay for continual learning Yang et al. (2023); Gu et al. (2024), critical data transfer in federated learning Huang et al. (2024a), and privacy preservation Dong et al. (2022).

Dataset distillation optimizes the synthetic data itself, with the objective of minimizing the training-aware information gap between the synthetic and original datasets. Broadly, dataset distillation can be decomposed into two complementary modules: distillation algorithms and data parameterization. The former defines how training-aware key information, such as training gradients Zhao et al. (2021), model parameters Cazenavette et al. (2022), or distributional statistics Zhao & Bilen (2023), is extracted, while the latter determines the representation format of the synthetic data. Although extensive research has focused on innovating distillation algorithms, exploration of the representation space has been relatively limited, with most approaches directly optimizing in the pixel space. Conventional methods typically adopt a naïve pixel grid as the parameterization scheme. Such dense representations fail to capture the relative importance of pixels and lead to substantial storage redundancy. Decoder-based parameterizations, in contrast, can leverage structural priors to enhance image realism and share common information, but they inevitably restrict the optimization space

and introduce additional computational and memory overhead. More recently, approaches based on implicit neural representations (INRs), such as DDiF Shin et al. (2025), have substantially reduced per-image storage and thereby improved the diversity of distilled datasets. However, their inherent per-pixel query decoding mechanism incurs prohibitive computational costs, particularly in high-resolution or large-batch scenarios, creating a severe performance bottleneck.

In this paper, we propose a new paradigm for dataset distillation, termed Gaussian Splatting Dataset Distillation (GSDD). Instead of relying on dense pixel grids, GSDD represents each distilled image with a sparse set of 2D Gaussians. Each Gaussian explicitly encodes training-aware image features that span multiple pixels, and optimizes only a few parameters such as position and shape, thereby replacing large-scale pixel-level optimization. This sparse representation significantly reduces the storage cost per image, which in turn increases the diversity of the distilled dataset and improves coverage across samples of varying difficulty. To further ensure the scalability of GSDD, we adapt a highly parallelized differentiable rasterizer that enables instantaneous high-quality rendering from parameters to images.

Our contributions can be summarized as follows:

- We propose a novel dataset parameterization framework based on 2D Gaussian representations. This is the first work that introduces sparse Gaussian representations into dataset distillation. The method is both simple and effective, and it can enhance the performance of a wide range of distillation algorithms.
- We provide a complete design of the Gaussian representation, including parameterization, optimization objectives, and an efficient batch-parallel Gaussian rasterization operator. This design ensures fast convergence, high distillation quality, and stable optimization.
- Extensive experiments demonstrate that our method achieves state-of-the-art performance on CIFAR-10, CIFAR-100, and multiple ImageNet subsets, while offering faster computation and lower memory consumption compared with prior approaches.

## 2 BACKGROUND AND MOTIVATION

### 2.1 RELATED WORK

**Dataset Distillation Algorithms** Modern deep learning models typically rely on large-scale training datasets, whereas the goal of dataset distillation is to replace the dataset with a much smaller set of high-quality training samples Wang et al. (2018). Unlike data selection, dataset distillation explicitly optimizes the synthetic data to achieve superior performance.

Formally, let the original dataset be denoted as

$$\mathcal{T} = \{(t_i, y_i) \mid i = 1, \cdots, N_{\mathcal{T}}\} \subseteq \mathcal{D},$$

where $\mathcal{D}$ represents the underlying data distribution. The distilled dataset (also referred to as the synthetic dataset) is defined as

$$\mathcal{S} = \{(s_j, y_j) \mid j = 1, \cdots, N_{\mathcal{S}}\},$$

where $t_i$ and $s_j$ denote original and distilled images, and $y_i$ and $y_j$ are their corresponding labels. Since this work adopts one-hot label encoding, without loss of generality we denote the distilled dataset as

$$\mathcal{S} = \{s_j \mid j = 1, \cdots, N_{\mathcal{S}}\}.$$

The storage footprint of $\mathcal{S}$ is typically required to be far smaller than that of $\mathcal{T}$. In dataset distillation, the synthetic dataset is treated as a set of learnable objects, where the pixel values of distilled images $s_j$ are regarded as optimization parameters. The ultimate goal is to train a model $\theta_{\mathcal{S}}$ on the compact distilled dataset such that its performance closely matches that of a model $\theta_{\mathcal{T}}$ trained on the full dataset Wang et al. (2018):

$$\mathcal{S} = \arg\min_{\mathcal{S}} \left| l(\theta_{\mathcal{S}}, \mathcal{D}) - l(\theta_{\mathcal{T}}, \mathcal{D}) \right|, \quad \theta_{\mathcal{S}} = \arg\min_{\theta} l(\theta, \mathcal{S}), \quad \theta_{\mathcal{T}} = \arg\min_{\theta} l(\theta, \mathcal{T}).$$

This optimization objective can be approached via meta-learning, but such methods often consume substantial GPU memory Feng et al. (2024) and yield suboptimal performance due to weak supervision signals. To address these challenges, many alternative distillation objectives have been proposed. The core idea is to extract and align critical training-aware information between $\mathcal{S}$ and $\mathcal{T}$. Mainstream approaches include trajectory matching(TM) Cazenavette et al. (2022); Du et al. (2023a); Liu et al. (2025); Guo et al. (2024); Cui et al. (2023), which constrains the distance between model parameters trained on the two datasets; distribution matching(DM) Zhao & Bilen (2023); Zhao et al. (2023); Liu et al. (2023); Wei et al. (2024); Wang et al. (2025b), which aligns feature distributions; and gradient matching(DC) Zhao et al. (2021); Lee et al. (2022); Du et al. (2023b), which constrains the gradients derived from the two datasets. Despite the drastic reduction in storage requirements, distilled datasets often achieve competitive training performance, and the resulting synthetic images can diverge significantly from the originals. Consequently, dataset distillation has found applications in continual learning Yang et al. (2023); Gu et al. (2024), privacy preservation Dong et al. (2022); Chung et al. (2024); Zheng et al. (2025), federated learning Huang et al. (2024a); Jia et al. (2025); Yan et al. (2025), and neural architecture search Such et al. (2020).

**Parameterization of Dataset Distillation** The parameterization of the distilled dataset $\mathcal{S}$ critically influences both performance and efficiency. Early methods directly optimized RGB pixels, but this dense representation wastes storage by ignoring spatial redundancy. To address this, decoder-based approaches Liu et al. (2022); Liu & Wang (2023); Cazenavette et al. (2023) learn latent codes with a shared decoder, which reduces redundancy but limits flexibility in optimization and adds computation costs. Dictionary-based methods Deng & Russakovsky (2022); Wei et al. (2023) instead combine basis images with index matrices to support domain adaptation and generalization, though they still store pixel-level bases and are restricted by linear combinations.

Recent studies have introduced compact low-level representations to reduce image redundancy and increase distilled samples within a fixed storage budget, such as downscaling Kim et al. (2022), frequency-domain transforms Shin et al. (2023), and implicit neural representations (INRs) Shin et al. (2025). INR-based methods show strong performance by encoding each image as a neural field with fewer parameters, but their efficiency is limited since decoding requires per-pixel RGB queries, causing heavy overhead for high-resolution and large-batch training.

**Gaussian Splatting** Gaussian splatting Kerbl et al. (2023) represents 3D scenes using Gaussian ellipsoids, offering higher rendering efficiency and explicit geometric control compared to NeRF-based representations Mildenhall et al. (2021). Originally developed for novel view synthesis, it initializes a set of Gaussian primitives within a 3D scene, which are then rendered from multiple viewpoints and optimized to match the corresponding ground-truth images, ultimately enabling arbitrary-view synthesis. In recent years, Gaussian splatting has been extended to a wide range of domains, including digital humans Qian et al. (2024); Zielonka et al. (2025), autonomous driving Zhou et al. (2024); Huang et al. (2024b), dynamic scene modeling Wu et al. (2024); Huang et al. (2024c), and 3D editing Chen et al. (2024). More recently, Gaussian representations have also been applied to 2D vision tasks such as image super-resolution Hu et al. (2024); Chen et al. (2025), image representation Zhu et al. (2025); Weiss & Bradley (2024); Zhang et al. (2025), and video representation Wang et al. (2025a), where Gaussian primitives are used to efficiently encode high-fidelity image of videos with fine-grained textures.

This work is the first to introduce Gaussian Splatting as a parameterization for dataset distillation, which leverages Gaussian representations as a simple yet efficient parameterization for distilled datasets. Unlike prior approaches that rely on network structures or complex operations, GSDD employs a CUDA-based differentiable rasterization algorithm to render sparse Gaussian primitives. As an explicit representation, it offers high flexibility in optimization, substantially increases dataset diversity under the same storage budget, and significantly accelerates both encoding and decoding. Rather than emphasizing fine-grained image fidelity, our method parameterizes entire distilled datasets with Gaussian representations and directly optimizes them for model training performance.

## 2.2 MOTIVATION

Gaussian representations enable faster decoding and rendering, especially compared to the state-of-the-art DDiF Shin et al. (2025) method. DDiF relies on implicit neural representations using a

SIREN network Sitzmann et al. (2020) to model distilled images, where each pixel's RGB value must be queried individually, as illustrated in Figure 1a. This approach incurs substantial time and memory overhead as the number and resolution of distilled images increase. In contrast, GSDD implements a custom CUDA operator for batched rendering of Gaussians. This allows all Gaussians to be efficiently rendered into distilled images in a single pass. As shown in Figure 1b, this design significantly reduces both inference and optimization time and memory usage, thereby improving the scalability of dataset distillation methods.

Gaussian representations enable compact modeling of individual distilled images with very few parameters. Traditional dataset distillation methods operate in the pixel space, where each pixel is assigned the same storage budget and optimized equally. This uniform treatment introduces unnecessary storage and computation costs. In contrast, GSDD can use a single Gaussian to represent a larger spatial area, covering multiple pixels simultaneously. This not only provides a more efficient image representation but also enhances optimization efficiency. As shown in Figure 1c, pixel-based methods optimize each pixel independently, which inevitably introduces redundancy and noise. In comparison, Gaussian-based optimization aggregates

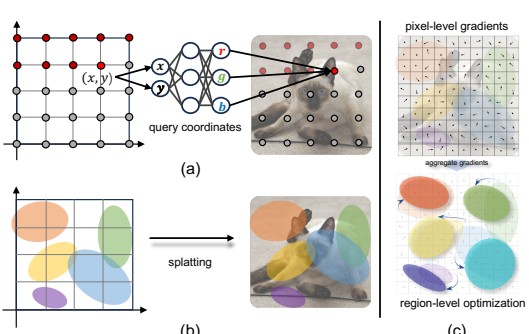

Figure 1: Comparison of Different Distilled Image Representation

pixel-level gradients within the support region of each Gaussian, resulting in more robust updates. Furthermore, each Gaussian can perform localized optimization by adjusting a small set of parameters such as position, color, and shape. This region-level update strategy improves both efficiency and effectiveness during training.

## 3 METHODOLOGY

### 3.1 PARAMETERIZATION OF DATASET DISTILLATION BY GAUSSIAN SPLATTING

We adopt a 2D Gaussian mixture model to parameterize distilled images. Each distilled image $s_j$ is represented as a set of $M$ Gaussian components, denoted by

$$G_j = \{g_k \mid k = 1, \cdots, M\}. \tag{1}$$

Specifically, for a pixel located at coordinates $(x, y)$, the contribution from the $k$-th Gaussian component $g_k$ is defined by the following unnormalized Gaussian function:

$$g_k(x, y; \mu_k, \Sigma_k) = \exp\left(-\tfrac{1}{2}(\mathbf{x} - \mu_k)^T \Sigma_k^{-1}(\mathbf{x} - \mu_k)\right), \tag{2}$$

where $\mathbf{x} = [x, y]^T$ denotes the pixel center and $\mu_k = [u_k, v_k]^T$ is the Gaussian mean. The covariance matrix $\Sigma_k$ determines the size, shape, and orientation of the Gaussian. To ensure that $\Sigma_k$ remains positive semi-definite during optimization, we parameterize it via a stable Cholesky decomposition, i.e., $\Sigma_k = L_k L_k^T$, where the lower-triangular matrix $L_k$ is given by

$$L_k = \begin{bmatrix} l_{11}^k & 0 \\ l_{21}^k & l_{22}^k \end{bmatrix}. \tag{3}$$

Each Gaussian $g_k$ is associated with a color vector $\mathbf{c}_k \in \mathbb{R}^3$ and an opacity $\alpha_k$. The final color of pixel $(x, y)$ is then synthesized by aggregating the contributions of all Gaussians:

$$\mathbf{c}(x, y) = \sum_{k=1}^{M} \alpha_k \cdot g_k(x, y; u_k, v_k, \Sigma_k) \cdot \mathbf{c}_k. \tag{4}$$

Thus, each Gaussian component can be fully described by a 9-dimensional vector $p_k = (u_k, v_k, l_{11}^k, l_{21}^k, l_{22}^k, \mathbf{c}_k, \alpha_k) \in \mathbb{R}^9$. Accordingly, the parameter set of an entire distilled image $s_j$ is

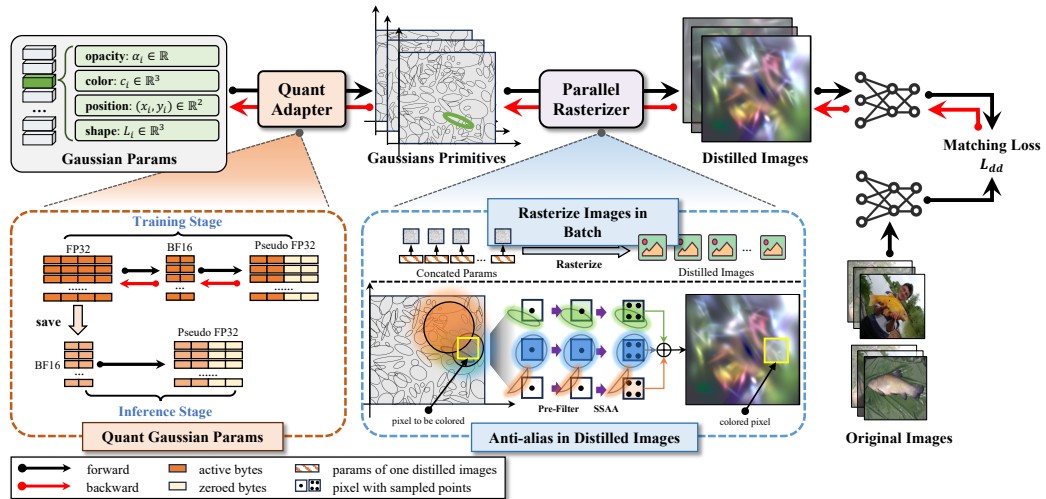

Figure 2: Overview of the proposed framework. Each Gaussian is parameterized by a total of 9 floating-point values. A single distilled image is represented by a set of Gaussians. During training, we first quantize the parameters to bf16 precision to obtain quantized Gaussian primitives. These are then rendered into distilled images using a customized rasterizer. During the rasterization process, we concatenate all primitives from the distilled dataset and feed them into a single batched rasterization kernel. This design enables efficient rendering and facilitates a compact data structure, as the entire distilled dataset can be managed by initializing only one object instance. To further improve rendering quality when Gaussians are sparse, we incorporate prefiltering and SuperSampling-based Anti-Aliasing techniques. These enhancements enable more accurate estimation of RGB values at each pixel. The rendered distilled images are then aligned with the original images for information matching, and the resulting gradients are backpropagated to update the Gaussian parameters.

given by $\mathbf{p}^j = \{p_k^j\}_{k=1}^M$, and the parameterization of the whole distilled dataset $\mathcal{S}$ is $\mathcal{P}_\mathcal{S} = \{\mathbf{p}^j\}_{j=1}^{N_\mathcal{S}}$. Based on this parameterization, $R$ can be seen as a rendering function, which maps the parameters $\mathbf{p}^j$ onto a predefined pixel coordinate grid $\mathcal{C}$, and is implemented as the cuda operator,

$$s_j = R(\mathbf{p}^j; \mathcal{C}), \tag{5}$$

which maps the parameters $\mathbf{p}^j$ onto a predefined pixel coordinate grid $\mathcal{C}$. Here, $\mathcal{C}$ denotes the set of all pixel coordinates, i.e.,

$$\mathcal{C} = \{x \mid x \in 1/W, \cdots, (W-1)/W\} \times \{y \mid y \in 1/H, \cdots, (H-1)/H\}. \tag{6}$$

This Gaussian-based parameterization is orthogonal to the choice of dataset distillation algorithm. Once a differentiable synthesis function is defined, it can be seamlessly integrated into any distillation framework. The optimization objective is to find the optimal distilled parameters $\text{Param}(\mathcal{S})^*$ that minimize the distillation loss $L_{\text{distill}}$:

$$\text{Param}(\mathcal{S})^* = \arg\min_{\text{Param}(\mathcal{S})} L_{\text{distill}}(\mathcal{S}, \mathcal{T}), \tag{7}$$

where $L_{\text{distill}}$ can correspond to any distillation objective.

To accelerate convergence and provide a good initialization for the optimization process, we pre-initialize the parameters of the distilled dataset. Specifically, we randomly sample a small subset of real images $\mathcal{S}_0, |\mathcal{S}_0| = N_\mathcal{S}$ from the original dataset $\mathcal{T}$, and minimize the mean squared error (MSE) between the synthesized images and these real samples. The initialization objective is thus:

$$\text{Param}(\mathcal{S})_0 = \arg\min_{\text{Param}(\mathcal{S})} L_{\text{MSE}}(\mathcal{S}, \mathcal{S}_0). \tag{8}$$

## 3.2 EFFICIENT GAUSSIAN SPLATTING DESIGN

The core of the above optimization lies in the differentiable renderer $R(\mathbf{p}^j; \mathcal{C})$, which maps Gaussian primitives to distilled images with both high visual fidelity and computational scalability. To enable efficient processing of Gaussian representations for large-scale dataset distillation, we incorporate

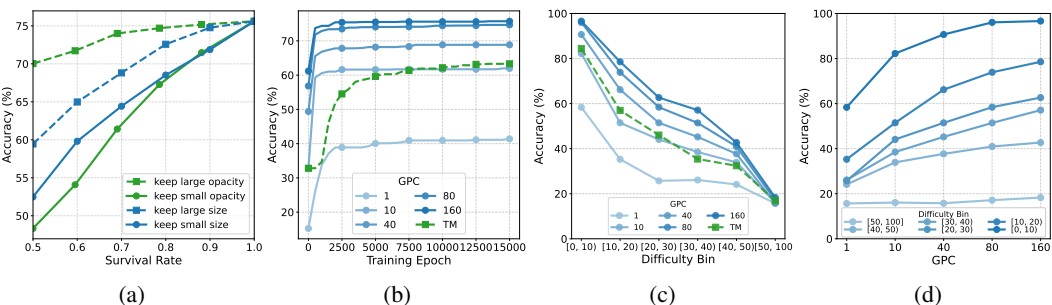

(a)    (b)    (c)    (d)

Figure 3: (a) Distillation performance under different Gaussian pruning strategies as a function of the remaining Gaussian ratio; (b) Test accuracy across training epochs with different GPC (Gaussian Images Per Class) under the same storage budget; (c) Test accuracy of the distilled dataset on samples of varying difficulty under the same storage budget; (d) Relationship between prediction accuracy on samples of different difficulty and GPC under the same storage budget. For fair comparison, TM is initialized with real images and serves as a baseline that represents pixel-based distillation.

several critical designs into the rendering pipeline, including parallel rendering of multiple distilled images, anti-aliasing strategies, spatial constraints on Gaussian positions and quantization.

Existing open-source rasterization operators are primarily designed for 3D scene reconstruction or single-image tasks such as representation learning and super-resolution, and generally lack support for multi-image rendering. To fully leverage the parallelism of modern GPUs, we implement customized data structures and rendering kernels for both the forward and backward passes. Specifically, all Gaussian primitives across the entire distilled dataset are represented as a single contiguous 1D vector, and a globally unique ID (GUID) system is used to assign threads that rasterize multiple images in parallel. Details are provided in Appendix C, and code will be open-sourced.

Some Gaussians may become highly anisotropic during optimization to capture high-frequency features such as edges, which necessitates anti-aliasing strategies. We adopt two complementary techniques: (i) analytic pre-filtering, which approximates each Gaussian's integral over a pixel area by convolving it with a unit pixel box filter, resulting in a modified covariance

$$\Sigma'_k = \Sigma_k + \Sigma_{box}, \quad \text{where} \quad \Sigma_{box} = \text{diag}\left(\tfrac{1}{12}, \tfrac{1}{12}\right), \tag{9}$$

introducing a minimum rendering variance that suppresses aliasing for extremely narrow Gaussians; and (ii) 2×2 supersampling anti-aliasing (SSAA), which averages multiple samples within each pixel area to smooth boundaries and reduce high-frequency artifacts with minimal overhead.

During optimization, some Gaussian centers tend to drift outside the normalized coordinate space $[-1, 1]^2$, where they receive no gradients and become unrecoverable, a phenomenon we term *Gaussian Escape*. To alleviate this, we apply a boundary regularization loss to slightly encourage Gaussian centers to remain within the viewable region:

$$l_{\text{boundary}} = -\mathbb{E}_k\left[\log(1 - \tilde{x}^2) + \log(1 - \tilde{y}^2)\right]. \tag{10}$$

In practice, the positions of Gaussian primitives are defined within the range $[-1, 1]$, and we perform coordinate transformations inside the CUDA operator. At each iteration, we clip the Gaussian positions to ensure they remain within the valid bounds:

$$\tilde{\mu}_k = \text{clip}(\mu_k, -1 + \epsilon, 1 - \epsilon). \tag{11}$$

We observe that Gaussian parameters exhibit robustness to reduced numerical precision, as the rasterization process does not suffer from cumulative precision errors. Based on this, we store all Gaussian parameters in bfloat16 (bf16) precision. During training, parameters are maintained in fp32 to ensure accurate updates, while bf16 casting is applied during the forward pass to allow the model to adapt to quantization effects. After training, all parameters are quantized to bf16.

### 3.3 ANALYSIS OF THE GAUSSIAN REPRESENTATION

**Sparsity improves optimization**    A key advantage of using Gaussian representations for image distillation lies in their ability to efficiently capture and encode training-aware image features that

span multiple pixels, using only a single or very few Gaussian primitives. As a result, this approach yields a more compact and efficient representation for each distilled image. We design a pruning experiment based on the following hypothesis: Gaussian components with larger spatial extent and higher opacity are more likely to carry critical training information. As shown in Figure 3a, when these large and opaque Gaussians are pruned first, the model performance drops sharply as the pruning ratio increases. In contrast, removing smaller and more transparent Gaussians has less effect on the performance. The results indicate that large Gaussian components not only cover broader pixel regions but also contribute more significantly to the final performance of the distilled dataset. This further confirms that the proposed Gaussian representation captures a highly sparse and information-dense encoding of the distilled image.

The Gaussian representation enables efficient modeling of distilled images with a small number of parameters. Its core mechanism lies in the ability to apply structured and coherent modifications to an entire pixel region by adjusting the properties of a single Gaussian component, such as its position, shape, color, and opacity. During backpropagation, each Gaussian also naturally aggregates the gradients of all pixels it covers. This optimization paradigm, which operates at the level of geometric primitives, offers improved robustness and efficiency compared to traditional pixel-wise optimiza-

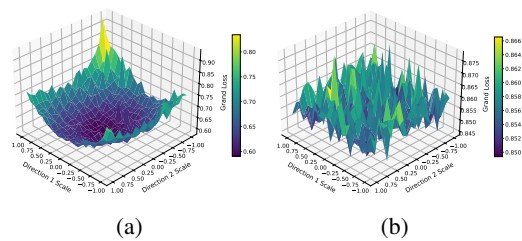

Figure 4: (a) Loss landscape of GSDD; (b) Loss landscape of pixel-based representation.

tion. As shown by the convergence curves in Figure 3b, GSDD achieves faster convergence and higher final performance than pixel-based methods. To further examine the underlying differences in optimization behavior, we visualize the loss landscapes of both approaches in Figure 4a and 4b. GSDD exhibits a smoother loss surface with a lower minimum and a wide basin resembling that of convex optimization, which substantially reduces the difficulty of gradient descent. In contrast, the pixel-based method presents a highly rugged and noisy loss landscape, making the optimization process more susceptible to poor local minima.

**Diversity improves scalability**   The parameter efficiency of Gaussian representations allows more distilled images to be synthesized under a fixed storage budget, thereby improving coverage of and generalization to more challenging samples. Prior work has shown that models often rely on memorization to generalize to hard examples, a phenomenon that is difficult to address in conventional dataset distillation settings. To evaluate this advantage, we conduct the following experiment: we first construct a difficulty spectrum by counting the number of incorrect predictions made by 100 independently trained models for each test sample. We then fix the overall storage budget and compare the performance of student models trained on distilled datasets with different Gaussian images Per Class(GPC) values, measuring their accuracy across different difficulty ranges. As shown in Figures 3c and 3d, the results reveal a clear trend: increasing the number of representable distilled images initially yields substantial performance gains on easy samples. With higher GPC values, the models further improve their accuracy on harder examples. These findings suggest that GSDD enhances downstream generalization by increasing the diversity of distilled images, thereby covering a broader range of the difficulty spectrum.

## 4   EXPERIMENTS

### 4.1   EXPERIMENTAL RESULTS

**Datasets and Baselines**   We evaluate our method on standard datasets: CIFAR-10, CIFAR-100 (at $32 \times 32$ resolution), six ImageNet subsets (at $128 \times 128$ resolution and $256 \times 256$ resolution), and ImageNet-1K (at $224 \times 224$ resolution). The general pipeline of dataset distillation involves two stages: (1) generating a synthetic dataset, and (2) training a model from scratch on the synthetic data. The performance of the trained model on the real test set is used to assess the quality of the distilled dataset. We include a wide range of recent dataset distillation baselines, including TM Cazenavette et al. (2022), FRePo Zhou et al. (2022), IDC Kim et al. (2022), FreD Shin et al. (2023), HaBa Liu et al. (2022), RTP Deng & Russakovsky (2022), HMN Liu et al. (2022),

Table 1: Test accuracy on ImageNet subsets (128 × 128) with trajectory matching (TM).

|  | Subset | Nette | Woof | Fruit | Yellow | Meow | Squawk |
|---|---|---|---|---|---|---|---|
|  | Original | 87.4 | 67.0 | 63.9 | 84.4 | 66.7 | 87.5 |
| Input sized | TM (Vanilla) | 51.4 | 29.7 | 28.8 | 47.5 | 33.3 | 41.0 |
|  | FRePo | 48.1 | 29.7 | — | — | — | — |
| Static | IDC | 61.4 | 34.5 | 38.0 | 56.5 | 39.5 | 50.2 |
|  | FreD | 66.8 | 38.3 | 43.7 | 63.2 | 43.2 | 57.0 |
| Parameterized | HaBa | 51.9 | 32.4 | 34.7 | 50.4 | 36.9 | 41.9 |
|  | SPEED | 66.9 | 38.0 | 43.4 | 62.6 | 43.6 | 60.9 |
|  | NSD | 68.6 | 35.2 | 39.8 | 61.0 | 45.2 | 52.9 |
| Generative Prior | GLaD | 38.7 | 23.4 | 23.1 | — | 26.0 | 35.8 |
|  | H-GLaD | 45.4 | 28.3 | 25.6 | — | 29.6 | 39.7 |
| Function | DDiF | 72.0 | 42.9 | 48.2 | 69.0 | 47.4 | 67.0 |
| Primitive | GSDD | **76.4** | **46.4** | **51.2** | **72.4** | **53.8** | **73.2** |

Table 2: Test accuracy on ImageNet subsets (128 × 128) with gradient matching (DC) and distribution matching (DM).

| DC | ImageNet Subset (128×128) | | | | | | Avg |
|---|---|---|---|---|---|---|---|
|  | Nette | Woof | Fruit | Yellow | Meow | Squawk | |
| DC (Vanilla) | 34.2 | 22.5 | 21.0 | 37.1 | 22.0 | 32.0 | 28.1 |
| GLaD | 35.4 | 22.3 | 20.7 | — | 22.6 | 33.8 | 27.0 |
| H-GLaD | 36.9 | 24.0 | 22.4 | — | 24.1 | 35.3 | 28.5 |
| IDC | 45.4 | 25.5 | 26.8 | — | 25.3 | 34.6 | 31.5 |
| FreD | 49.1 | 26.1 | 30.0 | — | 28.7 | 39.7 | 34.7 |
| DDiF | 61.2 | 35.2 | 37.8 | — | 39.1 | 54.3 | 45.5 |
| GSDD | **70.2** | **42.4** | **47.8** | **64.0** | **43.2** | **69.4** | **56.2** |
| **DM** | | | | | | | |
| DM (Vanilla) | 30.4 | 20.7 | 20.4 | 36.0 | 20.1 | 26.6 | 25.7 |
| GLaD | 32.2 | 21.2 | 21.8 | — | 22.3 | 27.6 | 25.0 |
| H-GLaD | 34.8 | 23.9 | 24.4 | — | 24.2 | 29.5 | 27.4 |
| IDC | 48.3 | 27.0 | 29.9 | — | 30.5 | 38.8 | 34.9 |
| FreD | 56.2 | 31.0 | 33.4 | — | 33.3 | 42.7 | 39.3 |
| DDiF | 69.2 | 42.0 | 45.3 | — | 45.8 | 64.6 | 53.4 |
| GSDD | **74.6** | **43.4** | **52.0** | **69.4** | **48.2** | **73.6** | **60.2** |

SPEED Wei et al. (2023), NSD Yang et al. (2025), GLaD Cazenavette et al. (2023), H-GLaD Zhong et al. (2025), LD3M Moser et al. (2024) and DDiF Shin et al. (2025).

**Experimental Details**    We use TM Cazenavette et al. (2022), DM Zhao & Bilen (2023), and DC Zhao et al. (2021) as the underlying distillation algorithms in our experiments. All parameters of Gaussians are trained using the Adam optimizer, and the learning rate for Gaussian parameters is uniformly set to 0.001. For CIFAR datasets, we apply ZCA whitening as a standard preprocessing step. We fix the boundary loss weight to $\lambda_{boundary} = 0.1$. Experiments are conducted on NVIDIA V100 and RTX 4090 GPUs. To initialize the Gaussian representation, we randomly sample real images from the original dataset and fit them with Gaussians before entering the standard distillation loop. We provide complete implementation details and hyperparameters in the Appendix A.

**Storage Budget and Evaluation Protocol**    We evaluate performance under varying storage budgets, specified in IPC (images per class). Since our method utilizes multiple low-storage Gaussian images, we additionally report GPC (Gaussian Images per Class) in Appendix A. The number of Gaussians per image is computed as $\text{pts} = \frac{\text{res} \times \text{res} \times 3 \times \text{IPC} \times 2}{\text{GPC} \times 9}$, where the factor of 2 accounts for the use of bf16, and the denominator 9 reflects the number of parameters per Gaussian.

**Performance Comparison**    Through the integration of GSDD, our method consistently outperforms the pixel-based TM baseline and other dataset parameterization methods on six ImageNet subsets with IPC=1, as shown in Table 1. The results on the higher-resolution ImageNet subset (256×256) are provided in Table 3, where our method remains superior at higher resolutions. In addition, we also report experiments conducted at a lower resolution on CIFAR-10/100, with the results presented in Appendix Table 15. When compared to the state-of-the-art method DDiF, our approach achieves consistent improvements, demonstrating highly competitive performance. These results highlight the effectiveness of GSDD in enhancing the quality of distilled datasets. We further report results on ImageNet-1K(224 × 224) in Table 18 in Appendix B.4.

**Universality to Matching Objectives**    GSDD is designed to be compatible with a wide range of dataset distillation algorithms and can be directly integrated with them. We evaluate GSDD under two widely adopted loss paradigms: gradient matching (DC) and distribution matching (DM). Experimental results, as shown in Table 2, confirm that GSDD robustly enhances performance across different distillation objectives.

**Cross-Architecture Performance**    An essential property of dataset distillation is robustness across model architectures. The distilled data should retain high performance even when the evaluation model differs from the one used during distillation. To assess this, we synthesize data using a ConvNet and evaluate it on a variety of downstream architectures, including ResNet, VGG, AlexNet and ViT. The results, summarized in Table 4 and Table 16, demonstrate that the increased diversity introduced by GSDD leads to substantial improvements in cross-architecture generalization.

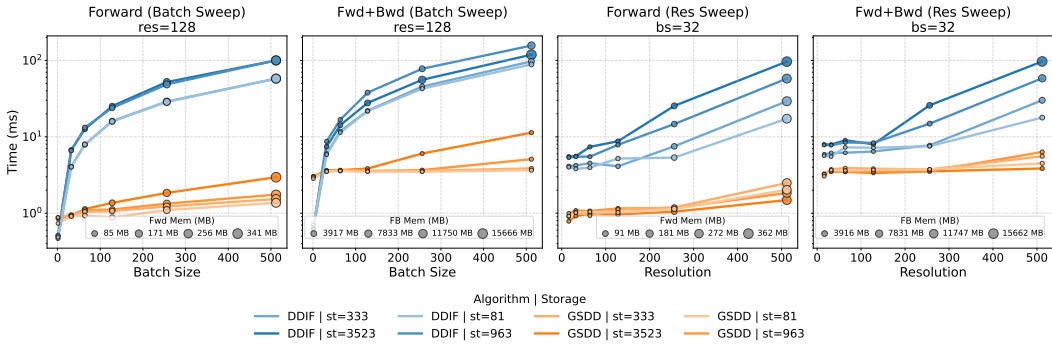

Figure 5: Performance comparison between GSDD and DDiF under the same per-image storage budget (abbreviated as *st* (floats)). Top row: Forward and forward+backward execution time and memory usage across varying image resolutions (with fixed batch size = 32). Bottom row: Same metrics across varying batch sizes (with fixed resolution = 128). GSDD consistently achieves lower latency and memory consumption, especially under high-resolution and large-batch scenarios.

Table 3: Results on ImageNet-subsets with Distribution Matching (DM) for $256 \times 256$

| Method | Nette | Woof | Fruit | Yellow | Meow | Squawk |
|---|---|---|---|---|---|---|
| Vanilla | 32.1 | 20.0 | 19.5 | 33.4 | 21.2 | 27.6 |
| IDC | 53.7 | 30.2 | 33.1 | 52.2 | 34.6 | 47.0 |
| FreD | 54.2 | 31.2 | 32.5 | 49.1 | 34.0 | 43.1 |
| LatentDD | 56.1 | 28.0 | 30.7 | — | 36.3 | 47.1 |
| DDiF | 67.8 | 39.6 | 43.2 | 63.1 | 44.8 | 67.0 |
| GSDD | **70.0** | **42.6** | **51.2** | **67.4** | **46.4** | **70.4** |

Table 4: Cross-Architecture Performance

| Subset | Nette | Woof | Fruit | Yellow | Meow | Squawk |
|---|---|---|---|---|---|---|
| TM | 22.0 | 14.8 | 17.1 | 22.3 | 16.2 | 25.5 |
| IDC | 27.9 | 19.5 | 23.9 | 28.0 | 19.8 | 29.9 |
| FreD | 36.2 | 23.7 | 23.6 | 31.2 | 19.1 | 37.4 |
| GLaD | 30.4 | 17.1 | 21.1 | — | 19.6 | 28.2 |
| H-GLaD | 30.8 | 17.4 | 21.5 | — | 20.1 | 28.8 |
| LD3M | 32.0 | 19.9 | 21.4 | — | 22.1 | 30.4 |
| DDiF | **59.3** | 34.1 | 39.3 | 51.1 | 33.8 | 54.0 |
| GSDD | 58.1 | **34.6** | **39.9** | **53.6** | **34.0** | **58.0** |

## 4.2 BENCHMARK RENDERING EFFICIENCY

To evaluate the efficiency and representational capacity of GSDD, we conduct an image-fitting experiment comparing several recent parameterization methods with the same storage budget, including FreD Shin et al. (2023), SPEED Wei et al. (2023) and DDiF Shin et al. (2025). Specifically, we use the ImageNette dataset and fit the first 100 images from each class (1,000 images in total) at a resolution of 128. The optimization is performed for 1,000 steps, and for each method we perform a grid search over learning rates and optimizers to ensure a fair comparison. As shown in Table 5, GSDD shows great performance across representational quality (PSNR), GPU memory usage, and runtime efficiency. High fidelity and low computational overhead of GSDD enables fast synthesis of more diverse (Higher GPC) distilled datasets, which in turn leads to improved downstream performance. We further verify in Section 4.3 that larger GPC values with fixed IPC generally correspond to stronger distilled data performance.

We further conduct a comprehensive quantitative comparison with the state-of-the-art DDiF method in terms of both decoding and training performance. As shown in Figure 5, we measure the inference time and memory consumption of both methods across varying per-image storage budgets, image resolutions, and batch sizes. The results demonstrate that GSDD consistently outperforms DDiF in both inference/training speed and memory efficiency. This advantage becomes especially pronounced when handling high-resolution inputs or large batch sizes, where GSDD achieves several-fold, or even an order-of-

Table 5: Comparison of Parameterization Methods on Image Fitting Efficiency and Representational Quality

| | FreD | SPEED | DDiF | GSDD |
|---|---|---|---|---|
| PSNR(dB)↑ | 15.69 | 17.77 | **23.23** | 23.22 |
| VRAM(MB)↓ | 280.4 | 298.5 | 1078.8 | **150.2** |
| Time(s)↓ | **15.5** | 69.4 | 1088.4 | 24.5 |

magnitude, reductions in both computation time and memory usage. These findings indicate that our method not only maintains strong image representation capabilities for distillation but also offers superior rendering speed and significantly lower memory footprint, thereby substantially improving the scalability of dataset distillation methods in practical deployment scenarios.

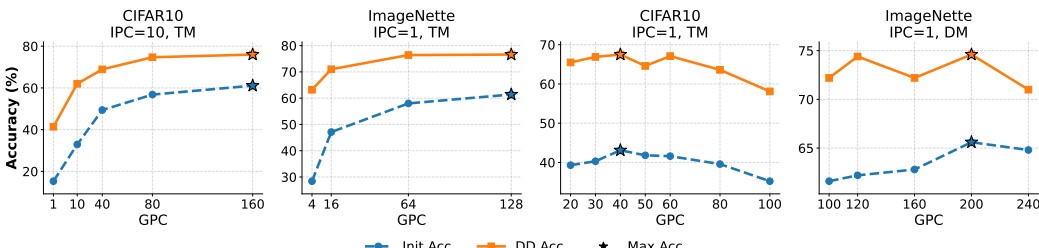

Figure 6: Effect of GPC (Gaussian Images Per Class) on distilled dataset performance across different datasets, storage budgets, and distillation algorithms (TM Cazenavette et al. (2022) and DM Zhao & Bilen (2023)). Initial Accuracy denotes the performance of initialized gaussian images (initialized on sampled real images). Distilled Accuracy denotes the performance after distillation.

### 4.3 ABLATION STUDY

To investigate the contribution of each component to model performance, we conduct a series of ablation studies, with results summarized in Table 6. The experiments show that opacity modeling and bf16 mixed-precision training are two major factors, as removing either leads to a notable performance drop. The boundary entry in the table refers to the result obtained when both hard clipping and loss regularization are removed. The impact of anti-aliasing exhibits a dependence on the dataset and the complexity of the representation. Its effect becomes important when the number of Gaussians used to represent each image is limited. Specifically, for datasets like CIFAR-10, where storage budgets are constrained and each image is represented by a small number of Gaussians, disabling anti-aliasing introduces rendering artifacts that degrade performance. In contrast, for datasets like ImageNette with denser Gaussians, aliasing artifacts are largely mitigated, and the performance gains from anti-aliasing become relatively minor.

Table 6: Ablation Study on Individual Components

| ImageNette | ACC |
|---|---|
| w/o all | 74.2 |
| w/o opacity | 74.8 |
| w/o boundary | 75.1 |
| w/o bf16 | 74.8 |
| w/o antialias | 76.4 |
| all | **76.4** |
| CIFAR-10 | ACC |
| w/o antialias | 74.5 |
| all | **75.5** |

We further examine the effect of GPC through an ablation study, as shown in Figure 6. When using the memory-efficient DM algorithm or when distilling under small storage budgets, performance increases with GPC at first but eventually declines. This is because small GPC values lead to insufficient diversity, whereas excessively large GPC values allocate too few Gaussians per image, weakening each image's representational capacity. In more memory-consuming settings, the optimal turning point of GPC becomes difficult to reach, and performance tends to improve monotonically as GPC increases. Figure 6 also shows that the performance of the initial dataset serves as a useful reference for estimating a reasonable GPC range.

### 5 CONCLUSION

This work introduces sparse Gaussian representations into the task of dataset distillation parameterization. We propose a complete framework for encoding distilled images using 2D Gaussian primitives, including the underlying mathematical formulation, a parallel differentiable renderer, and several tailored components to handle the sparsity of distilled datasets. Specifically, we incorporate anti-aliasing and spatial constraints to enhance the utility of Gaussians and improve overall distillation quality. Through extensive quantitative and qualitative experiments, we demonstrate that sparse Gaussian representations facilitate more effective optimization, offer strong scalability, and provide better coverage across samples of varying difficulty. As shown in our results, GSDD achieves highly competitive performance under various datasets and storage budgets, while also exhibiting superior computational efficiency compared to DDiF. The proposed method is simple, plug-and-play, and readily compatible with existing distillation algorithms, leaving ample room for future extensions. Future work may focus on scaling to larger datasets, extending Gaussian representations to video modalities, and exploring dynamic density control of Gaussian primitives to further enhance representation quality.

REPRODUCIBILITY STATEMENT

We provide a comprehensive description of our method in Section 3, and detailed hyperparameters, optimizers, and network architectures for each dataset in Appendix A. All experiments are conducted on publicly available datasets. Our core implementation has been submitted to the supplemental material and more detailed codebase will be open-sourced upon acceptance of the paper.

ETHICS STATEMENT

This work complies with the ICLR Code of Ethics. It does not involve any unethical experimentation, nor does it use private, sensitive, or personally identifiable data. All datasets employed in our experiments are publicly available and properly licensed. No data was collected from human subjects or required approval from an institutional review board.

Our research promotes responsible stewardship of machine learning technology. In particular, by advancing the use of synthetic representations for dataset distillation, our approach supports privacy-preserving research and reduces the reliance on real-world data that might raise privacy concerns.

We have taken care to ensure transparency and reproducibility. All implementation details, including hyperparameters, architectures, and code necessary for reproduction, are provided in the supplemental materials and will be open-sourced upon acceptance. Furthermore, we acknowledge and cite all prior works that our approach builds upon, respecting intellectual contributions in line with academic and ethical standards.

LLM USAGE STATEMENT

All research components presented in this paper, including the initial literature review, problem formulation, methodological development, and experimental design, were independently conceived and developed by the authors, without any contribution from large language models (LLMs).

LLMs were used solely as general-purpose tools for non-substantive assistance. Specifically, we employed LLMs to help polish the writing, check grammar, and clarify LaTeX syntax during the preparation of the manuscript. In all such cases, the authors critically reviewed, verified, and revised the generated content to ensure its accuracy and originality. The authors take full responsibility for the final text, including any parts informed by LLM assistance.

In addition, we made limited use of AI-assisted coding tools (e.g., GitHub Copilot in VSCode) for basic code autocompletion during code implementation. LLMs were also used to assist in initial framework scaffolding for code visualization and to suggest potential debugging strategies. All core algorithmic components and experimental pipelines were manually written, verified, and refined by the authors. Every LLM-assisted suggestion was critically assessed and edited to ensure correctness and alignment with the intended methodology.

LLMs were not used to generate, modify, or interpret any experimental results or conclusions.

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

# A  EXPERIMENTAL DETAILS

## A.1  DATASETS

We conduct experiments on eight datasets in total, including CIFAR-10, CIFAR-100, and six ImageNet subsets: *imagenette*, *imagewoof*, *imagefruit*, *imagemeow*, *imagesquawk*, and *imageyellow*. CIFAR-10 consists of 60,000 images of resolution $32 \times 32$ across 10 classes, with 50,000 training images and 10,000 test images. CIFAR-100 has the same resolution and number of images, but covers 100 classes. Each ImageNet subset contains images of resolution $128 \times 128$ from 10 classes, with more than 10,000 images per subset.

## A.2  NETWORK ARCHITECTURES

We adopt ConvNet architectures as the backbone networks. A Depth-$n$ ConvNet consists of $n$ blocks followed by a fully-connected layer, where each block is composed of a $3 \times 3$ convolutional layer with 128 filters, instance normalization, a ReLU activation, and $2 \times 2$ average pooling with stride 2. Specifically, we use ConvNetD3 for CIFAR-10 and CIFAR-100, and ConvNetD5 for the ImageNet subsets.

## A.3  IMAGE INITIALIZATION

For the initialization of synthetic images, all experiments adopt the mean squared error (MSE) loss and use the Adam optimizer with a learning rate of $1 \times 10^{-2}$.

## A.4  DATASET DISTILLATION

During dataset distillation, Gaussian points are optimized with the Adam optimizer and a learning rate of $1 \times 10^{-2}$. The number of optimization iterations is set to 15,000 unless otherwise specified.

Table 7: Hyperparameters for dataset distillation on CIFAR-10 $32 \times 32$.

| Setting | GPC | num_points | syn_steps | max_start_epoch | expert_epochs | lr_lr | lr_init | batch_syn | zca |
|---|---|---|---|---|---|---|---|---|---|
| IPC=1 | 30 | 22 | 60 | 30 | 2 | 1e-5 | 1e-2 | 300 | True |
| IPC=10 | 160 | 42 | 60 | 30 | 2 | 1e-5 | 1e-2 | 380 | True |
| IPC=50 | 250 | 136 | 60 | 30 | 2 | 1e-5 | 1e-2 | 360 | True |

Table 8: Hyperparameters for dataset distillation on CIFAR-100 $32 \times 32$.

| Setting | GPC | num_points | syn_steps | max_start_epoch | expert_epochs | lr_lr | lr_init | batch_syn | zca |
|---|---|---|---|---|---|---|---|---|---|
| IPC=1 | 30 | 22 | 60 | 30 | 2 | 1e-5 | 1e-2 | 512 | True |
| IPC=10 | 80 | 85 | 60 | 30 | 2 | 1e-5 | 1e-2 | 512 | True |
| IPC=50 | 400 | 85 | 60 | 30 | 2 | 1e-5 | 1e-2 | 640 | True |

Table 9: Hyperparameters for dataset distillation on ImageNet-subset TM $128 \times 128$.

| Setting | GPC | num_points | syn_steps | max_start_epoch | expert_epochs | lr_lr | lr_init | batch_syn | zca |
|---|---|---|---|---|---|---|---|---|---|
| IPC=1 | 64 | 170 | 20 | 40 | 2 | 1e-6 | 1e-2 | 150 | False |
| IPC=10 | 640 | 170 | 20 | 40 | 2 | 1e-5 | 1e-2 | 160 | False |

## A.5  PERFORMANCE BENCHMARKING

In Table 5, we evaluate the image-fitting performance of FreD Shin et al. (2023), SPEED Wei et al. (2023), DDiF Shin et al. (2025), and GSDD. We select 100 images from each class of the ImageNette dataset, resulting in a total of 1,000 images, all at a resolution of 128. Each method is constrained to the same storage budget, meaning that, on average, each image is represented using 963 floating-point parameters. For SPEED, we follow the parameter-counting formula provided in its original

Table 10: Hyperparameters for dataset distillation on ImageNet-subset $128 \times 128$ (DM).

| Setting | GPC | num_points | batch_real | batch_syn | zca | Iteration |
|---------|-----|-----------|-----------|----------|-----|-----------|
| IPC=1 | 200 | 54 | 1024 | 2000 | False | 20000 |

Table 11: Hyperparameters for dataset distillation on ImageNet-subset $256 \times 256$ (DM).

| Setting | GPC | num_points | batch_real | batch_syn | zca | Iteration |
|---------|-----|-----------|-----------|----------|-----|-----------|
| IPC=1 | 120 | 364 | 512 | 512 | False | 20000 |

Table 12: Hyperparameters for dataset distillation on ImageNet-subset $128 \times 128$ (DC).

| Setting | GPC | num_points | batch_real | batch_syn | zca | Iteration |
|---------|-----|-----------|-----------|----------|-----|-----------|
| IPC=1 | 200 | 54 | 720 | 2000 | False | 5000 |

Table 13: Performance gains of individual components on ImageNette.

| w/ none | w/ boundary | w/ bf16 | w/ opacity | w/ antilias |
|---------|-------------|---------|-----------|-------------|
| 74.2 | 74.6 | 76.0 | 76.0 | 74.4 |

Table 14: Fine-grained ablation on boundary constraints. 'reg' denotes the boundary loss regularization defined in Eq. 10, and 'clip' denotes the clipping operation defined in Eq. 11.

| w/o reg | w/o clip | w/o reg+clip | w/ reg+clip |
|---------|----------|--------------|-------------|
| 76.2 | 76.2 | 75.1 | 76.4 |

paper,

$$\#\text{Params} = DK + 1.5NHk + R(3D^2 2 + 7D) + L(D + 1),$$

and, given the storage constraint, we solve for all feasible hyperparameter combinations. The final configuration used in our experiments is (D=32, N=32, k=135, R=2, L=64). We attempted to evaluate HaBa Liu et al. (2022) as well, but its model structure cannot be solved inversely under the resolution of 128 and the storage budget of 963 parameters. FreD, DDiF, and GSDD exhibit clear advantages in terms of estimating and meeting the storage budget. For each method, we search over Adam and SGD optimizers and learning rates spaced exponentially by a factor of 10, and we report the highest PSNR achieved. All models are trained for 1,000 iterations using MSE loss. Each experiment is repeated three times with random initialization, and the average result is reported. The results demonstrate that our method achieves strong representational capability and computational efficiency.

# B  MORE EXPERIMENTAL RESULTS

## B.1  PERFORMANCE COMPARISON ON LOW-DIMENSIONAL DATASETS

As shown in Table 15, our method GSDD consistently achieves the best performance across both CIFAR-10 and CIFAR-100 under all IPC settings. Notably, GSDD surpasses all baselines by a clear margin in the low-data regime (IPC=1), achieving 67.6% on CIFAR-10 and 43.0% on CIFAR-100. In high-data settings (IPC=50), GSDD maintains its superiority, reaching 77.7% and 53.1% respectively. These results demonstrate the strong generalization ability and scalability of our primitive-based distillation framework.

## B.2  DETAILED CROSS-ARCHITECTURE RESULTS

We evaluate the generalization ability of distilled data across different architectures, including AlexNet, VGG11, ResNet18, and ViT. The network implementations follow those provided

Table 15: Classification accuracy on CIFAR-10 and CIFAR-100 under different IPC settings.

| | Method | CIFAR10 | | | CIFAR100 | | |
|---|---|---|---|---|---|---|---|
| | | IPC=1 | IPC=10 | IPC=50 | IPC=1 | IPC=10 | IPC=50 |
| Input sized | TM | 46.3±0.8 | 65.3±0.7 | 71.6±0.2 | 24.3±0.3 | 40.1±0.4 | 47.7±0.2 |
| | FRePo | 46.8±0.7 | 65.5±0.4 | 71.7±0.2 | 28.7±0.1 | 42.5±0.2 | 44.3±0.2 |
| Static | IDC | 50.0±0.4 | 67.5±0.5 | 74.5±0.2 | — | — | — |
| | FreD | 60.6±0.8 | 70.3±0.3 | 75.8±0.1 | 34.6±0.4 | 42.7±0.2 | 47.8±0.1 |
| Parameterized | HaBa | 48.3±0.8 | 69.9±0.4 | 74.0±0.2 | — | — | 47.0±0.2 |
| | RTP | 66.4±0.4 | 71.2±0.4 | 73.6±0.5 | 34.0±0.4 | 42.9±0.7 | — |
| | HMN | 65.7±0.3 | 73.7±0.1 | 76.9±0.2 | 36.3±0.2 | 45.4±0.2 | 48.5±0.2 |
| | SPEED | 63.2±0.1 | 73.5±0.2 | 77.7±0.4 | 40.4±0.4 | 45.9±0.3 | 49.1±0.2 |
| | NSD | **68.5±0.8** | 73.4±0.2 | 75.2±0.6 | 36.5±0.3 | 46.1±0.2 | — |
| Function | DDiF | 66.5±0.4 | 74.0±0.4 | 77.5±0.3 | 42.1±0.2 | 46.0±0.2 | 49.9±0.2 |
| Primitive | GSDD | 67.6±0.4 | **75.5±0.3** | **77.7±0.5** | **43.0±0.1** | **47.4±0.3** | **53.1±0.2** |

in Cazenavette et al. (2023). For training, the learning rate is set to $1 \times 10^{-3}$ for AlexNet, $5 \times 10^{-5}$ for ViT, and $1 \times 10^{-2}$ for the remaining architectures. All models are optimized using Adam with a cosine annealing learning rate schedule. We report the results averaged over three independent runs, as shown in Figure 16.

Table 16: Detailed cross-architecture performance comparison.

| Test Net | Method | Nette | Woof | Fruit | Yellow | Meow | Squawk |
|---|---|---|---|---|---|---|---|
| AlexNet | TM | 13.2 ± 0.6 | 10.0 ± 0.0 | 10.0 ± 0.0 | 11.0 ± 0.2 | 9.8 ± 0.0 | — |
| | IDC | 17.4 ± 0.9 | 16.5 ± 0.7 | 17.9 ± 0.7 | 20.6 ± 0.9 | 16.8 ± 0.5 | 20.7 ± 1.0 |
| | FreD | 35.7 ± 0.4 | 23.9 ± 0.7 | 15.8 ± 0.7 | 19.8 ± 1.2 | 14.4 ± 0.5 | 36.3 ± 0.3 |
| | DDiF | 60.7 ± 2.3 | **36.4 ± 2.3** | 41.8 ± 0.6 | **56.2 ± 0.8** | **40.3 ± 1.9** | **60.5 ± 0.4** |
| | GSDD | **63.1 ± 1.3** | 33.6 ± 0.9 | **41.9 ± 2.0** | 53.5 ± 0.9 | 38.1 ± 0.8 | 60.1 ± 0.2 |
| VGG11 | TM | 17.4 ± 2.1 | 12.6 ± 1.8 | 11.8 ± 1.0 | 16.9 ± 1.1 | 13.8 ± 1.3 | — |
| | IDC | 19.6 ± 1.5 | 16.0 ± 2.1 | 13.8 ± 1.3 | 16.8 ± 3.5 | 13.1 ± 2.0 | 19.1 ± 1.2 |
| | FreD | 21.8 ± 2.9 | 17.1 1.7 | 12.6 ± 2.6 | 18.2 ± 1.1 | 13.2 ± 1.9 | 18.6 ± 2.3 |
| | DDiF | 53.6 ± 1.5 | 29.9 ± 1.9 | 33.8 ± 1.9 | 44.2 ± 1.7 | **32.0 ± 1.8** | 37.9 ± 1.5 |
| | GSDD | **56.8 ± 1.6** | **32.7 ± 1.4** | **34.1 ± 3.0** | **57.3 ± 4.5** | 31.3 ± 1.8 | **55.9 ± 3.4** |
| ResNet18 | TM | 34.9 ± 2.3 | 20.7 ± 1.0 | 23.1 ± 1.5 | 43.4 ± 1.1 | 22.8 ± 2.2 | — |
| | IDC | 43.6 ± 1.3 | 23.2 ± 0.8 | 32.9 ± 2.8 | 44.2 ± 3.5 | 28.2 ± 0.5 | 47.8 ± 1.9 |
| | FreD | 48.8 ± 1.8 | 28.4 ± 0.6 | 34.0 ± 1.9 | 49.3 ± 1.1 | 29.0 ± 1.8 | 50.2 ± 0.8 |
| | DDiF | **63.8 ± 1.8** | 37.5 ± 1.9 | 42.0 ± 1.9 | **55.9 ± 1.0** | 35.8 ± 1.8 | **62.6 ± 1.5** |
| | GSDD | 59.1 ± 3.0 | **39.5 ± 2.0** | **42.6 ± 0.8** | **55.9 ± 1.2** | **40.9 ± 2.9** | **62.6 ± 0.6** |
| ViT | TM | 22.6 ± 1.1 | 15.9 ± 0.4 | 23.3 ± 0.4 | 18.1 ± 1.3 | 18.6 ± 0.9 | — |
| | IDC | 31.0 ± 0.6 | 22.4 ± 0.8 | 31.1 ± 0.8 | 30.3 ± 0.6 | 21.4 ± 0.7 | 32.2 ± 1.2 |
| | FreD | 38.4 ± 0.7 | 25.4 ± 1.7 | 31.9 ± 1.4 | 37.6 ± 2.0 | 19.7 ± 0.8 | 44.4 ± 1.0 |
| | DDiF | **59.0 ± 0.4** | **32.8 ± 0.8** | 39.4 ± 0.8 | **47.9 ± 0.6** | **27.0 ± 0.6** | **54.8 ± 1.1** |
| | GSDD | 53.4 ± 0.6 | 32.6 ± 1.1 | **41.0 ± 1.2** | 47.6 ± 0.8 | 25.8 ± 0.4 | 53.5 ± 2.1 |

## B.3 MORE ABLATION STUDY

**Different initialization strategy** Prior methods often initialize distilled data using real images, which may introduce additional privacy risks. We further conduct an ablation study on different initialization strategies, including random initialization. However, we find that optimizing Gaussian representations from purely random initialization is inherently unstable: the initial Gaussian points are extremely small and sparsely distributed, creating large gaps that prevent effective gradient propagation. To mitigate this issue, we introduce a Solid-Color Warmup strategy. Instead of fitting any

real images, all Gaussian representations are first optimized to fit a single solid-color image. This process reduces point sparsity and adjusts the initial point sizes, after which standard distillation begins. As shown in Table 17, even without real-image initialization, our warmup-based strategy achieves strong results and surpasses several methods that depend on real-image initialization.

**More ablation study on individual components** We first provide an ablation study on the 'reg' and 'clip' operations in the boundary constraints to supplement the boundary components reported in Table 6. As shown in the results, both operations contribute to alleviating the Gaussian escape problem. In addition to Table 6, we further perform an incremental ablation experiment starting from a baseline with no techniques, and adding each technique individually. Table 13 shows that bf16 and opacity contribute the most to the final performance, whereas boundary and antialiasing offer smaller gains. Similar trends can also be observed in Table 6.

Table 17: Initialization Strategy Ablation Results

|      | Init   | Acc. |
| ---- | ------ | ---- |
| IDC  | real   | 61.4 |
| FreD | real   | 66.8 |
| NSD  | random | 68.6 |
| DDiF | real   | 72.0 |
| GSDD | random | 63.6 |
| GSDD | warmup | 69.4 |
| GSDD | real   | 76.4 |

### B.4 RESULTS ON IMAGENET-1K

Enabled by the efficiency of recent dataset-distillation frameworks, such as SRe2L Yin et al. (2023) and RDED Sun et al. (2024), dataset parameterization at the ImageNet-1K Deng et al. (2009) scale has become feasible for the first time. In this work, we build upon SRe2L and evaluate our method under both IPC=1 and IPC=10 settings. In addition, we reproduce the IDC Kim et al. (2022) baseline, where each image is spatially downscaled by a factor of $\sqrt{k}$, resulting in $k$-times more synthetic samples.

For IDC, under IPC=1 we report the best-performing $k$ selected from $k \in \{4, 20, 50\}$, and under IPC=10 from $k \in \{4, 10, 20\}$. As shown in Table 18, our method improves the performance of SRe2L on ImageNet-1K, particularly in the IPC=1 setting. Although the gains diminish at IPC=10, our approach still maintains a clear margin over IDC.

Table 18: Performance on ImageNet-1K.

|           | IPC=1 | IPC=10 |
| --------- | ----- | ------ |
| SRe2L     | 0.9   | 21.3   |
| SRe2L+IDC | 13.4  | 41.5   |
| SRe2L+GSDD | 29.0  | 46.2   |

Table 19: Hyperparameters for each method on ImageNet-1K.

|            | IPC=1   | IPC=10   |
| ---------- | ------- | -------- |
| SRe2L+IDC  | k=20    | k=5      |
| SRe2L+GSDD | GPC=50  | GPC=200  |

### B.5 RESULTS ON CROSS-RESOLUTION GENERALIZATION

Gaussian Splatting naturally supports rendering at arbitrary resolutions through supersampling, enabling us to distill data at a low resolution (e.g., 128) while still using it to train models at a higher resolution (e.g., 256). This effectively reduces the computation when distilling datasets at large resolutions. Following results and setup in DDiF Shin et al. (2025), we conduct cross-resolution experiments as shown in Table 21.

## C PARALLELIZATION STRATEGY FOR BATCHED RENDERING

To scale the rendering pipeline from a single image to an entire batch of images while maximizing GPU throughput, we designed a global ID system coupled with corresponding data structure management. The core idea is to consolidate the rendering of multiple independent images into a single, massive computational task.

### C.1 CORE CHALLENGE AND THE GLOBAL ID SYSTEM

The parallelization strategy for batched rendering evolves the original single-image pipeline's local indexing system into a global, batch-aware framework. The original approach operated in a local context where screen tiles were indexed from 0 to M-1 for a single image, and data structures pertained only to that instance.

To scale this process across a batch of B images, we introduced a Globally Unique ID (GUID) system. This system creates a single, contiguous address space for all tiles across the entire batch. A local tile j from image i is re-indexed into a global tile id using the linear transformation:

$$\text{global\_tile\_id} = i \times M + j \tag{12}$$

This re-indexing prompted adaptations to our key CUDA kernels. The intersection mapping kernel was modified to calculate this global tile id for each generated intersection record, enabling the subsequent sorting operation to correctly group tasks on a global, batch-wide level. Correspondingly, the rasterization kernel is now launched with a 1D grid of B×M thread blocks, where each block's index directly corresponds to a global tile id. Inside the kernel, this global ID is decomposed back into its constituent image id and local tile id. This allows each thread block to precisely identify which tile of which image it is responsible for, ensuring it writes the final pixel color to the correct memory offset within the batch output tensor.

To support this parallelization strategy, we further employ specific data structures at the Python (PyTorch) level to manage and transfer data efficiently.

## C.2 1D FLATTENING AND CONTIGUOUS MEMORY LAYOUT

We avoid using Python lists or non-contiguous tensors to store the parameters of different images. Instead, all Gaussian parameters (e.g., means, features, and Cholesky components) across the batch are concatenated and stored in a single large contiguous PyTorch tensor. For a batch of $B$ images with $N$ points each, the means tensor has a shape of $(B \times N, 2)$, rather than being represented as a list of $B$ tensors of shape $(N, 2)$. This design ensures memory contiguity and eliminates the performance overhead caused by frequent concatenation.

By combining the global ID system at the CUDA kernel level with a contiguous memory layout at the framework level, we have constructed an end-to-end rendering architecture that achieves high throughput for large-scale batched rendering tasks.

## C.3 BENCHMARK ACROSS DIFFERENT GPU ARCHITECTURES

To evaluate the stability of our CUDA-based rasterizer, we conduct systematic benchmarking on five different GPU architectures. We use an image-fitting task in which Gaussian representations reconstruct 1000 images sampled from ImageNet (10 classes × 100 images in total). Each experiment is repeated three times, and we average all results. We test four storage budgets (21, 170, 682, 2730 Gaussians per image). All experiments used PyTorch 2.5.1 and CUDA ≤ 12.0.

Table 20: Runtime, memory usage, and PSNR of our rasterizer benchmarked across five GPU architectures under varying storage budgets.

| GPU Model | Time (s) | | | | Memory (GB) | | | | PSNR | | | |
|---|---|---|---|---|---|---|---|---|---|---|---|---|
| | 21 | 170 | 682 | 2730 | 21 | 170 | 682 | 2730 | 21 | 170 | 682 | 2730 |
| Tesla-V100-SXM2-32GB | 23.65 | 26.11 | 30.28 | 42.94 | 1.66 | 1.70 | 1.81 | 2.28 | 20.24 | 25.93 | 30.65 | 36.09 |
| NVIDIA-A100-SXM4-40GB | 10.45 | 12.87 | 19.09 | 41.53 | 1.66 | 1.70 | 1.81 | 2.28 | 20.24 | 26.03 | 31.06 | 36.08 |
| NVIDIA-GeForce-RTX-3090 | 10.33 | 16.07 | 25.44 | 61.38 | 1.66 | 1.70 | 1.81 | 2.28 | 20.24 | 26.03 | 31.06 | 36.08 |
| NVIDIA-GeForce-RTX-4090 | 7.60 | 9.38 | 14.06 | 31.00 | 1.66 | 1.70 | 1.81 | 2.28 | 20.24 | 26.02 | 31.03 | 36.06 |
| NVIDIA-GeForce-RTX-5090 | 7.13 | 6.99 | 9.54 | 21.79 | 1.66 | 1.70 | 1.81 | 2.28 | 20.23 | 26.02 | 31.03 | 36.06 |

As shown in Table 20. Different GPUs exhibit different runtimes due to heterogeneous compute capabilities, and RTX5090 is the fastest, while RTX3090 is the slowest. Memory usage remained identical across all platforms, and PSNR variation is small across architectures. Slight performance drops on V100 are expected due to older hardware, but results on A100/3090/4090/5090 remain highly consistent.

For IDC, SPEED, and FreD, we adopt their best-performing upsampling strategies. For DDiF and GSDD, supersampling is applied, allowing lossless upscaling of the distilled images. GSDD maintains competitive performance in cross-resolution tasks. Although its relative performance drop is larger than DDiF, which is likely due to GSDD's inherent sparsity, reducing its ability to capture fine-grained features when evaluated at higher resolutions.

Table 21: Test accuracies with different resolutions and networks. Images are first distilled with low resolution(128) and test at higher resolution(256/512).

| test resolution | test network | method | accuracy↑ | difference↓ | ratio↓ |
|---|---|---|---|---|---|
| 256 | ConvNetD5 | Vanilla | 31.2 | 20.2 | 0.39 |
| | | IDC | 55.0 | 6.4 | 0.10 |
| | | SPEED | 58.8 | 8.1 | 0.12 |
| | | FreD | 56.4 | 10.4 | 0.16 |
| | | DDiF | 66.3 | **5.7** | **0.08** |
| | | GSDD | **66.4** | 10.0 | 0.13 |
| | ConvNetD6 | Vanilla | 44.0 | 7.3 | 0.14 |
| | | IDC | 55.4 | 6.0 | 0.10 |
| | | SPEED | 62.6 | 4.3 | 0.06 |
| | | FreD | 61.8 | 5.0 | 0.07 |
| | | DDiF | 70.6 | **1.4** | **0.02** |
| | | GSDD | **71.6** | 4.8 | 0.06 |
| 512 | ConvNetD5 | Vanilla | 27.4 | 24.0 | 0.47 |
| | | IDC | 39.5 | 21.9 | 0.36 |
| | | SPEED | 45.0 | 21.9 | 0.33 |
| | | FreD | 42.9 | 23.9 | 0.36 |
| | | DDiF | 58.7 | **13.3** | **0.18** |
| | | GSDD | **59.4** | 17.0 | 0.22 |
| | ConvNetD6 | Vanilla | 41.2 | 10.1 | 0.20 |
| | | IDC | 51.5 | 9.9 | 0.16 |
| | | SPEED | 60.1 | 6.8 | 0.10 |
| | | FreD | 56.3 | 10.5 | 0.16 |
| | | DDiF | **69.0** | **3.0** | **0.04** |
| | | GSDD | 67.8 | 8.6 | 0.11 |

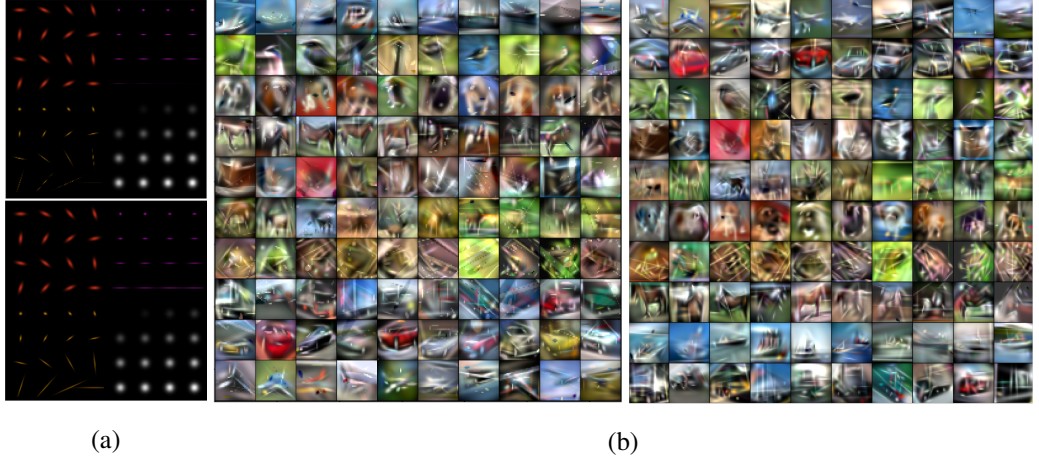

(a)            (b)

Figure 7: Comparison between aliasing and anti-aliasing. (a) shows the schematic illustration, while (b) demonstrates the effect on CIFAR-10 dataset visualization. The anti-aliasing strategy leads to smoother and clearer patterns.

## D    VISUALIZATION OF ANTIALIAS AND GAUSSIAN ESCAPE

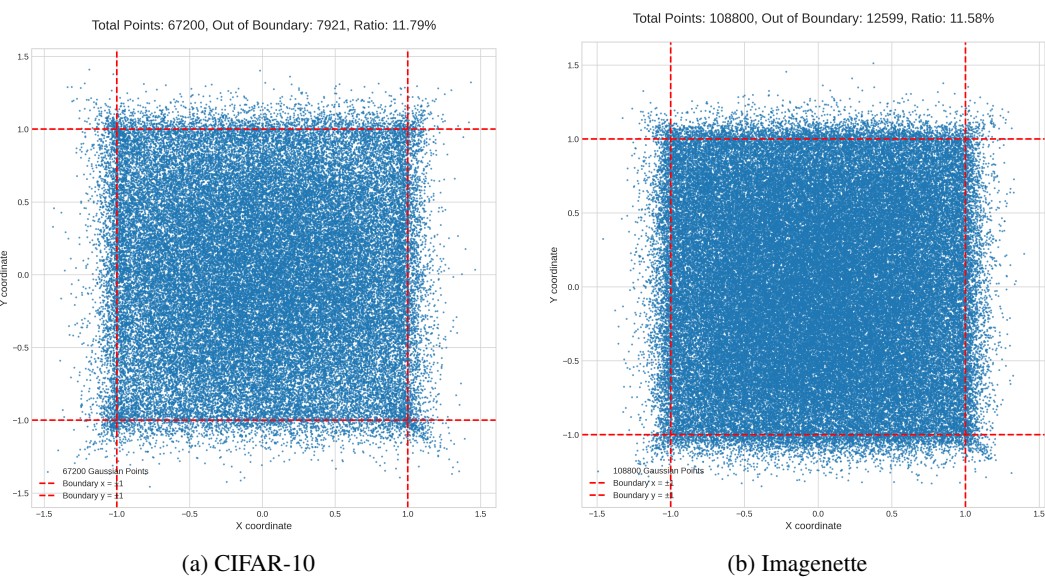

(a) CIFAR-10          (b) Imagenette

Figure 8: Visualization of Gaussian point distributions without boundary constraints. We plot the center coordinates of all Gaussian points after training and observe clear escaping phenomena on both CIFAR-10 and Imagenette, which reduces the representational efficiency of Gaussian points.

Figure 7 visualizes the test images without and with anti-aliasing, along with the actual renderings on CIFAR-10. It is evident that when the Gaussian points are relatively elongated, the rendering quality degrades, resulting in pronounced aliasing artifacts and discontinuous stripes, which in turn adversely affect the performance of the distilled dataset.

In Figure 8, we visualize the escaping phenomenon of Gaussian points after training when no boundary constraints are applied. We plot the distributions of all Gaussian point centers and observe that, on both CIFAR-10 and Imagenette datasets, Gaussian points tend to escape beyond the valid range, which in turn undermines their representational efficiency.

# E VISUALIZATION OF GSDD

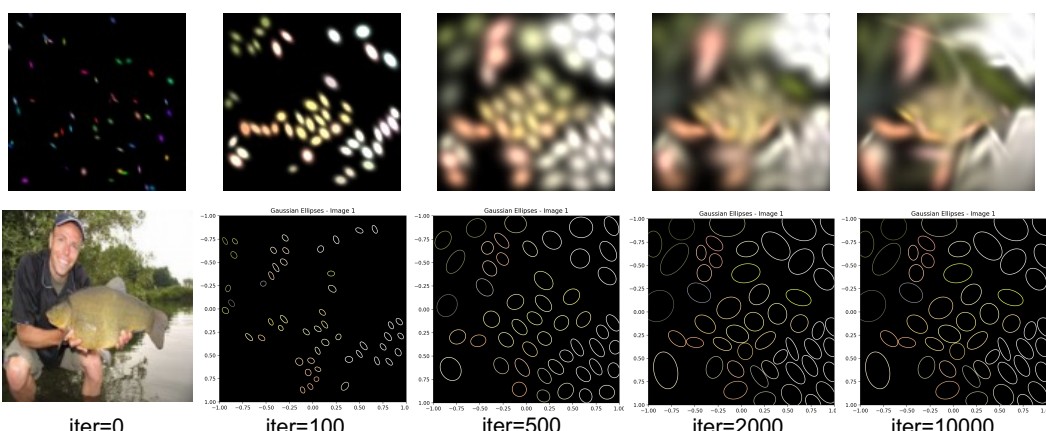

iter=0    iter=100    iter=500    iter=2000    iter=10000

Figure 9: Visualization of the Gaussian initialization process. The first row shows the actual renderings of Gaussian points at different iterations, while the second row represents each Gaussian as an ellipse encoding its position, shape, and color. During optimization, Gaussian points continuously adjust their positions, shapes, and colors to fit the target image.

We visualize the process of Gaussian initialization in Figure 9. The first row illustrates the actual rendering results of Gaussian points, while the second row depicts the Gaussians as ellipses that encode their positions, shapes, and colors. It can be observed that, during optimization, Gaussian points continuously move, reshape, and adjust their colors to fit the target image.

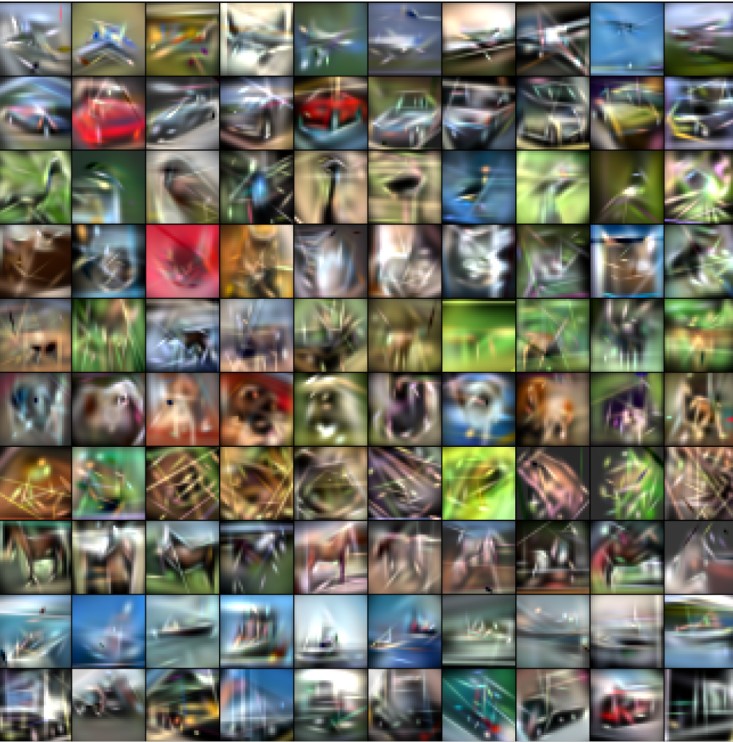

Figure 10: Visualization of Distilled Images on CIFAR-10

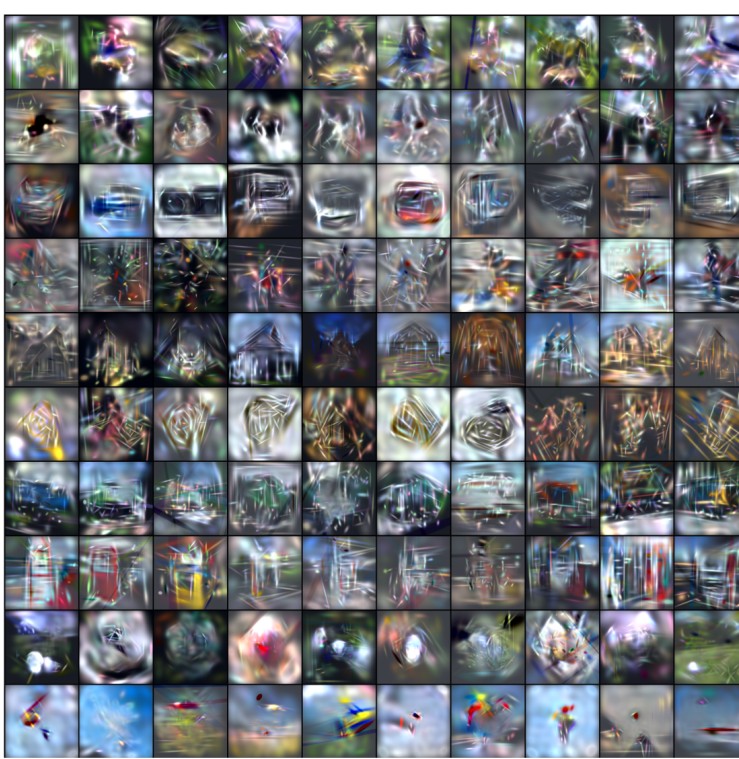

Figure 11: Visualization of Distilled Images on CIFAR-100

Figure 12: Visualization of Distilled Images on ImageNette

