# OpenReview forum: "Beyond Pixels: Efficient Dataset Distillation via Sparse Gaussian Representation"
_ICLR.cc/2026/Conference — Submitted to ICLR 2026_

### Official Review · Reviewer_zWtU · 2025-10-27

**Soundness:** 3
**Presentation:** 2
**Contribution:** 2
**Rating:** 4
**Confidence:** 3

**Summary:**

The paper proposes GSDD, a dataset distillation method using a sparse 2D Gaussian representation to replace dense pixels and costly implicit neural representations, aiming to enhance efficiency and scalability. According to the reported results, it achieves notable speed and memory gains over INR-based baselines (e.g., DDiF) while maintaining comparable or better performance across different DD algorithms and architectures.

**Strengths:**

1. This is the first work to introduce a parametric Gaussian framework into the field of dataset distillation.
2. The method employs customized CUDA operators to significantly improve computational efficiency.

**Weaknesses:**

**Major:**

1. The authors emphasize in the introduction and motivation that DDiF struggles with larger and higher-resolution datasets, yet the experiments in this paper do not substantiate this claim. Specifically, DDiF is evaluated on the ImageNet subset with a resolution of 256×256, while GSDD experiments are limited to 128×128 or even 32×32. To convincingly demonstrate GSDD’s superior generalization, it should at least show improvements on datasets of comparable scale and resolution.
2. The method section provides extensive background on parametric Gaussian modeling but lacks theoretical justification for why Gaussian functions are inherently more effective than INRs. There is no formal reasoning or analysis showing that 2D Gaussian bases offer intrinsic advantages over other differentiable bases (e.g., wavelet or trigonometric) for gradient matching in dataset distillation. As a result, the choice of Gaussian appears heuristic and engineering-driven rather than grounded in theoretical insight.
3. The paper does not theoretically formalize the superiority of this representation. Most arguments focus on engineering aspects such as CUDA-based rasterization being faster than neural network querying, rather than proving that sparse explicit primitives are fundamentally more suitable for encoding discriminative knowledge than implicit continuous functions.
4. The paper lacks sensitivity or ablation studies on key hyperparameters, particularly the number of Gaussian primitives per image and their initialization strategy. Since these parameters critically affect performance, the absence of such analysis makes it difficult to assess the robustness and practical applicability of the method.
5. The paper claims to evaluate cross-architecture transfer from ConvNet to ResNet and VGG. However, Table 4 only shows the performance of different DD methods on ImageNet subsets and does not present any cross-architecture results, so the claimed generalization across architectures is not supported.

**Minor:**

1. The notation $R$ in Equation (5) is undefined.
2. The reference “Sequential subset matching for dataset distillation” has inconsistent author formatting compared to other entries.

**Questions:**

See weakness.

---

> ### Author Response · Authors · 2025-11-19
>
> We sincerely thank the reviewer for your valuable suggestions. In response, we have conducted new experiments on 256-resolution ImageNet subsets, evaluated the representational ability of GSDD, demonstrated how efficiency boosts distillation performance, added ablation studies, and revised the writing (including appendix, symbols, and citation).
>
> > **W1:**  To convincingly demonstrate GSDD’s superior generalization, it should at least show improvements on datasets of comparable scale and resolution.
>
> We agree that distillation at higher resolutions is particularly demanding. Below we report results on **ImageNet-Subset at 256×256**, using DM as the baseline distillation algorithm:
>
> Table 1: Results on ImageNet-Subsets (256x256)
>
> | |Nette|Woof|Fruit|Yellow|Meow|Squawk|
> | - | - | - | - | - | - | - |
> |Vanilla|32.1|20.0|19.5|33.4|21.2|27.6|
> |IDC| 53.7|30.2|33.1|52.2|34.6|47.0|
> |FreD|54.2|31.2|32.5|49.1|34.0|43.1|
> |LatentDD|56.1|28.0|30.7|-| 36.3|47.1|
> |DDiF|67.8|39.6|43.2|63.1|44.8|67.0|
> |GSDD|**70.0**|**42.6**|**51.2**|**67.4**| **46.4** | **70.4** |
>
> On 256-resolution ImageNet-Subset, GSDD consistently outperforms all baselines by a clear margin.
>
> We further apply GSDD to **ImageNet-1K**, using SRe2L as the baseline. We additionally evaluate SRe2L with IDC, where IDC downsamples each image by a factor of k, thereby increasing the total number of images by k times.
>
> Table 2: Results on ImageNet-1K
>
> | | IPC=1| IPC=10|
> |-|-|-|
> |SRe2L|0.9|21.3|
> |SRe2L+IDC|13.4(k=20)|41.5(k=4)|
> |SRe2L+GSDD|29.0(GPC=50)|46.2(GPC=200)|
>
> Best results for IDC are selected from $k \in \lbrace 4,20,50 \rbrace$ for IPC=1, and $k\in \lbrace 4,10,20\rbrace$ for IPC=10.
>
> It can be seen that our method boosts the performance of SRe2L, especially for the IPC=1 setting. It is worth noting that this is the first time for parameterization method in DD applied on ImageNet-1K, thanks to efficient distillation framework like SRe2L, as well as the efficiency and representativeness of GSDD.
>
> > **W2:** Lack theoretical justification for why Gaussian functions are inherently more effective than INRs. There is no formal reasoning or analysis showing that 2D Gaussian bases offer intrinsic advantages over other differentiable bases.
>
> Richer and higher-quality parameterizations tend to yield better distilled datasets[1-2]. DDiF[2] demonstrates the strong representational ability of INRs through reconstruction experiments. We agree that it is important to justify the representational ability of 2D Gaussian bases, either theoretically or empirically.
>
> From a theoretical viewpoint:
>
> - **FreD[1]** leverages the natural low-frequency bias of images and operates in the frequency domain, achieving compact representations. However, it (i) relies on a fixed pre-selection of frequency components; and (ii) only optimizes coefficients of these pre-chosen basis functions, while the basis itself is not learnable. This limits its expressiveness during distillation.
> - **DDiF[2]** can be regarded as a generalized form of FreD. In DDiF, both frequency and phase are learnable, effectively making the basis functions themselves trainable. This provides stronger representational flexibility.
>
> From the perspective of differentiable bases, **Gaussian splatting** offers a similar decomposition: we can view each Gaussian’s **opacity** as a learnable coefficient and the Gaussian kernel itself as a learnable 2D basis function. In GSDD, both the coefficients (opacities) and the bases (Gaussian positions, scales, and shapes) are jointly optimized.
>
> Directly comparing the theoretical expressiveness of these function spaces is difficult. Therefore, we perform a **quantitative comparison of representational power and efficiency** across existing parameterization methods, including SPEED, FreD, DDiF, and GSDD.
>
> Concretely, we sample 100 images($128\times 128$) per class from ImageNette, yielding 1000 images in total, and allocate each parameterization the same **average storage budget of 963 parameters per image** (matching DDiF’s design for ImageNette, IPC = 1, DIPC = 51). We run 1000 optimization steps and tune the learning rate and schedule for each method.
>
> Table 3: Benchmark of representativeness and efficiency
>
> | |FreD|SPEED|DDiF| GSDD|
> | - | - | - | - | - |
> | PSNR(dB)↑ |15.69|17.77|**23.23** | 23.22|
> | GPU Memory(MB)↓ | 280.4| 298.5 |1078.8| **150.2** |
> | time(s)↓ | **15.5** | 69.4|1088.4| 24.5|
>
> We observe that:
>
> - GSDD and DDiF achieve similarly strong reconstruction quality (PSNR ≈ 23.2 dB).
> - GSDD, however, uses less memory and much less time than DDiF due to the efficiency of the rasterization operator.
>
> This mirrors the trend in 3D reconstruction: while implicit representations (e.g., NeRF-style INRs) may achieve very high rendering quality, Gaussian Splatting has seen broader adoption in practice because of its superior computational efficiency. In addition, explicit and implicit representations can be further combined to encode structural information and feature jointly.

---

> ### Author Response · Authors · 2025-11-19
>
> > **W3:** The paper does not theoretically formalize the superiority of this representation. This paper should prove that sparse explicit primitives are fundamentally more suitable for encoding discriminative knowledge than implicit continuous functions.
>
> While it is difficult (and perhaps too strong) to claim that **sparse explicit primitives are "fundamentally more suitable"** than implicit continuous functions in encoding discriminative knowledge, we could provide a more concrete argument that, **GSDD’s efficiency allows it to achieve substantially higher representation diversity (under the same storage budget), and this diversity is strongly correlated with improved distillation performance.**
>
> Dataset diversity is crucial for performance [3-6]. However, existing methods struggle to maximize diversity due to the trade-off between representation quality and efficiency, as shown reply to W2.
>
> Specifically, we evaluate the effect of **GPC (Gaussian Images Per Class)** under various datasets, distillation algorithms(TM[7] and DM[8]), and storage budgets:
>
> Table 4: CIFAR10，IPC=1, with TM
>
> |GPC|20|30|40|50|60|80|100|
> |-|-|-|-|-|-|-|-|
> | DD_acc | 65.5 | 66.9 | **67.5** | 64.6 | 67.1 | 63.6 | 58.1 |
>
> Table 5: CIFAR10，IPC=10, with TM
>
> | GPC| 1| 10| 40| 80| 160|
> | - | - | - | - | - | - |
> | DD_acc | 41.4 | 62.0 | 68.9 | 74.7 | **76.0** |
>
> Table 6: ImageNette, IPC=1, with TM
>
> | GPC| 4| 16| 64| 128|
> | - | - | - | - | - |
> | DD_acc | 63.2 | 71| 76.4 | **76.6** |
>
> Table 7: ImageNette, IPC=1, with DM
>
> | GPC    | 100  | 120  | 160  | 200| 240  |
> | - | - | - | - | - | - |
> | DD_acc | 72.2 | 74.4 | 72.2 | **74.6** | 71.0 |
>
> We observe that DD performance generally increases with larger GPC. Performance only begins to drop when the storage budget is extremely small and GPC becomes so large that the representational capacity is significantly weakened. GSDD benefits from highly efficient representations, allowing GPC to be expanded. This enables us to directly assess how increasing diversity impacts distilled dataset performance.
>
> This finding offers a interesting perspective, one can trade off per-image representation quality by adopting very sparse representations in order to dramatically increase overall diversity and thereby improve accuracy.
>
> > **W4**: The paper lacks sensitivity or ablation studies on key hyperparameters, particularly the number of Gaussian primitives per image and their initialization strategy.
>
> We conducte ablations on GPC, including experiments on CIFAR-10 (IPC=10) and ImageNette (IPC=1) under the TM algorithm, as well as ImageNette (IPC=1) under the DM algorithm. **The corresponding results have been presented in our response to W3.** Given a fixed IPC (Images Per Class) budget, GPC is inversely related to the number of Gaussian points per image: a higher GPC leads to fewer Gaussian points per image.
>
> We also provide a simple heuristic for estimating GPC, where the performance of the initial fitting stage can be used to estimate a reasonable range for the optimal GPC.
>
> `init_acc`: Performance of initialized gaussian images (initialized on sampled real images). `DD_acc`: Performance after distillation.
>
> Table 8: CIFAR10，IPC=1, with TM.
>
> | GPC| 20| 30| 40| 50| 60| 80 |100|
> | - | - | - | - | - | - |-|-|
> | init_acc | 39.3 | 40.3 | **43.1** | 41.8 | 41.6 |39.6|35.2|
> | DD_acc   | 65.5 | 66.9 | **67.5** | 64.6 | 67.1 |63.6|58.1|
>
> Table 9: ImageNette, IPC=1, with TM
>
> | GPC| 4| 16| 64| 128|
> | - | - | - | - | - |
> | init_acc | 28.4 | 71   | 58   | **61.4** |
> | DD_acc   | 63.2 | 71   | 76.4 | **76.6** |
>
> Table 10: ImageNette, IPC=1, with DM
>
> | GPC      | 100  | 120  | 160  | 200| 240  |
> | - | - | - | - | - | - |
> | init_acc | 61.6 | 62.2 | 62.8 | **65.6** | 64.8 |
> | DD_acc   | 72.2 | 74.4 | 72.2 | **74.6** | 71.0 |
>
> The optimal GPC range for `init_acc` and `DD_acc` is consistent. This is intuitive: increasing GPC reduces the number of Gaussian points per image under fixed storage, creating a trade-off between data quality and diversity that applies to both the initialization set and the distilled set.
>
> **Ablation on Initialization:** We found that optimizing Gaussian representations from random initialization is unstable. Random Gaussian points are initially too small and sparse (large gaps), preventing effective reception of gradient signals.
>
> To address this, we utilize a Solid Color Warmup. All Gaussian representations first fit a single solid color image to reduce gaps and adjust point size, followed by distillation. **This process does not fit any real images.**
>
> Table 11: Performance Comparison of Initialization Strategies
>
> | | IDC(w/ init) | FreD(w/ init) | NSD(w/o init) | DDiF(w/ init) | GSDD(w/o init) | GSDD(w/o init, warmup) | GSDD(w/ init) |
> | - | - | - | - | - | - | - | - |
> | imagenette | 61.4| 66.8| 68.6| 72.0| 63.6| 69.4| 76.4|
>
> **Even without real image initialization, our method (w/ warmup) achieves excellent results**, outperforming several methods that rely on real image initialization

---

> ### Author Response · Authors · 2025-11-19
>
> > **W5**: The paper claims to evaluate cross-architecture transfer from ConvNet to ResNet and VGG. However, Table 4 only shows the performance of different DD methods on ImageNet subsets and does not present any cross-architecture results, so the claimed generalization across architectures is not supported.
>
> **Summary:** We add explicit references in the main text to the appendix and provide more detailed per-architecture comparisons.
>
> We apologize for the missing explicit reference in the main text. In the appendix (Table 10 in older version, Table 14 in the latest manuscript), we have reported cross-architecture performance on four test networks, including **ResNet18, AlexNet, VGG11, and ViT**, across six ImageNet-Subset datasets (ImageNette, ImageWoof, etc.). We will clarify this in the main paper. In addition, Reviewer 7jwx suggested that we should add results with prior methods for each architecture individually. We therefore update results in Table 10 (Table 14 in the latest manuscript)  accordingly. The updated results are:
>
> | **Test Net** | **Method** | **Nette**| **Woof**| **Fruit**| **Yellow**| **Meow**| **Squawk**|
> | - | - | - | - | - | - | - | - |
> | **AlexNet**  | **TM**| 13.2 ± 0.6| 10.0 ± 0.0| 10.0 ± 0.0| 11.0 ± 0.2| 9.8 ± 0.0| - |
> | | **IDC**    | 17.4 ± 0.9| 16.5 ± 0.7| 17.9 ± 0.7| 20.6 ± 0.9| 16.8 ± 0.5     | 20.7 ± 1.0     |
> | | **FreD**   | 35.7 ± 0.4| 23.9 ± 0.7| 15.8 ± 0.7| 19.8 ± 1.2| 14.4 ± 0.5     | 36.3 ± 0.3     |
> | | **DDiF**   | 60.7 ± 2.3     | **36.4 ± 2.3** | 41.8 ± 0.6     | **56.2 ± 0.8** | **40.3 ± 1.9** | **60.5 ± 0.4** |
> | | **GSDD**   | **63.1 ± 1.3** | 33.6 ± 0.9     | **41.9 ± 2.0** | 53.5 ± 0.9     | 38.1 ± 0.8     | 60.1 ± 0.2     |
> | **VGG11**    | **TM**     | 17.4 ± 2.1     | 12.6 ± 1.8     | 11.8 ± 1.0     | 16.9 ± 1.1     | 13.8 ± 1.3     | - |
> | | **IDC**    | 19.6 ± 1.5     | 16.0 ± 2.1     | 13.8 ± 1.3     | 16.8 ± 3.5     | 13.1 ± 2.0     | 19.1 ± 1.2     |
> | | **FreD**   | 21.8 ± 2.9     | 17.1 1.7       | 12.6 ± 2.6     | 18.2 ± 1.1     | 13.2 ± 1.9     | 18.6 ± 2.3     |
> | | **DDiF**   | 53.6 ± 1.5     | 29.9 ± 1.9     | 33.8 ± 1.9     | 44.2 ± 1.7     | **32.0 ± 1.8** | 37.9 ± 1.5     |
> | | **GSDD**   | **56.8 ± 1.6** | **32.7 ± 1.4** | **34.1 ± 3.0** | **57.3 ± 4.5** | 31.3 ± 1.8     | **55.9 ± 3.4** |
> | **ResNet18** | **TM**     | 34.9 ± 2.3     | 20.7 ± 1.0     | 23.1 ± 1.5     | 43.4 ± 1.1     | 22.8 ± 2.2     | - |
> | | **IDC**    | 43.6 ± 1.3     | 23.2 ± 0.8     | 32.9 ± 2.8     | 44.2 ± 3.5     | 28.2 ± 0.5     | 47.8 ± 1.9     |
> | | **FreD**   | 48.8 1.8       | 28.4 ± 0.6     | 34.0 ± 1.9     | 49.3 ± 1.1     | 29.0 ± 1.8     | 50.2 ± 0.8     |
> | | **DDiF**   | **63.8 ± 1.8** | 37.5 ± 1.9     | 42.0 ± 1.9     | **55.9 ± 1.0** | 35.8 ± 1.8     | **62.6 ± 1.5** |
> | | **GSDD**   | 59.1 ± 3.0     | **39.5 ± 2.0** | **42.6 ± 0.8** | **55.9 ± 1.2** | **40.9 ± 2.9** | **62.6 ± 0.6** |
> | **ViT**| **TM**| 22.6 ± 1.1| 15.9 ± 0.4| 23.3 ± 0.4| 18.1 ± 1.3| 18.6 ± 0.9     | - |
> | | **IDC**| 31.0 ± 0.6| 22.4 ± 0.8| 31.1 ± 0.8| 30.3 ± 0.6| 21.4 ± 0.7| 32.2 ± 1.2     |
> | | **FreD**| 38.4 ± 0.7| 25.4 ± 1.7| 31.9 ± 1.4| 37.6 ± 2.0| 19.7 ± 0.8| 44.4 ± 1.0     |
> | | **DDiF**| **59.0 ± 0.4** | **32.8 ± 0.8** | 39.4 ± 0.8| **47.9 ± 0.6** | **27.0 ± 0.6** | **54.8 ± 1.1** |
> | | **GSDD**| 53.4 ± 0.6| 32.6 ± 1.1| **41.0 ± 1.2** | **47.6 ± 0.8** | 25.8 ± 0.4     | 53.5 ± 2.1     |
>
> We have updated Table 10 in the appendix and added an explicit reference to it in the main context.
>
> > Minor Weakness1: The notation $R$ in Equation (5) is undefined.
> >
> > Minor Weakness2: The reference “Sequential subset matching for dataset distillation” has inconsistent author formatting compared to other entries.
>
> **$R$ can be seen as a rendering function**, which maps the parameters $\mathbf{p}^j$  onto a predefined pixel coordinate grid $\mathcal{C}$, and is implemented as the cuda operator.
>
> Sorry for the inconsistent formatting, and we have corrected both the formatting and definition in our manuscript.
>
> Most of the results have already been updated in the latest manuscript, and in particular, our ablation study on GPC is presented with a more intuitive visualization in Figure 6.
>
> Thank you again for your valuable feedback, and please feel free to let us know if you have any further questions.
>
> References:
>
> [1] Frequency Domain-Based Dataset Distillation, NeurIPS, 2023
>
> [2] Distilling dataset into neural field, ICLR, 2025
>
> [3] On the Diversity and Realism of Distilled Dataset: An Efficient Dataset Distillation Paradigm, CVPR, 2024
>
> [4] Efficient Dataset Distillation via Minimax Diffusion, CVPR, 2204
>
> [5] Diversity-enhanced distribution alignment for dataset distillation, ICCV, 2025
>
> [6] DELT: a simple diversity-driven EarlyLate training for dataset distillation, CVPR, 2025
>
> [7] Dataset distillation by matching training trajectories, CVPR, 2022
>
> [8] Dataset Condensation With Distribution Matching, WACV, 2023

---

### Official Review · Reviewer_7jwx · 2025-10-31

**Soundness:** 3
**Presentation:** 3
**Contribution:** 3
**Rating:** 4
**Confidence:** 4

**Summary:**

This paper proposes GSDD to address the redundancy and poor scalability issues inherent in previous dataset distillation methods that rely on dense pixel-level representations. Specifically, GSDD introduces a sparse representation based on 2D Gaussian primitives to encode distilled images. Each Gaussian captures region-level discriminative information with only a few parameters optimization. This proposed Gaussian-based representation reduces storage overhead, increases dataset diversity and improves optimization stability. Also, to ensure efficiency and scalability, this paper implements a CUDA-based differentiable rasterizer for parallel rendering of multiple distilled images. Experiments demonstrate that GSDD achieves state-of-the-art performance on CIFAR-10, CIFAR-100, and ImageNet subsets with efficiency gains.

**Strengths:**

1. The paper is well written with clear and informative figures and well-motivated for essential efficiency issues in dataset distillation field.
2. The proposed GSDD is simple yet effective, with a novel and intuitive idea. The transformation from per-pixel to per-region representation is well motivated and directly addresses the inefficiency of dense pixel-level encoding.
3. The proposed GSDD can be integrated into other DD methods while improving the efficiency and performance.
4. The performance of the proposed method is good with less computational cost.

**Weaknesses:**

1. [major] The efficiency of the proposed method relies heavily on CUDA-based parallel rasterizer, which raises concerns about portability and reproducibility. Different GPU hardware-specific optimizations can lead to slightly different acceleration behaviors, the performance and results may vary between different GPU architectures or operator implementations.
2. [major] Since distilled images are generated from sets of Gaussian primitives rather than explicit pixels, this representation may have weaker semantic visualization and interpretability, potentially limiting the model generalization ability across different architectures. As shown in Tables 4 and 10, GSDD performance improvement across different architectures is relatively limited compared to performance on the same architecture. Furthermore, this paper only reports the average results for each architecture without providing detailed comparisons per architecture, making the cross-architecture performance results less convincing and verifiable.
3. [minor] While the proposed method demonstrates efficiency and scalability, experiments have primarily focused on relatively low-resolution datasets, such as subsets of CIFAR and ImageNet (low-resolution version). To better verify whether the method remains effective on high-resolution or more complex datasets, it could be extended to ImageNet full size with full resolution.

**Questions:**

1. As mentioned in weakness major 1, it would be helpful if the authors could provide some explanation or empirical evidence regarding how consistent the results are across different GPU architectures.
2. For weakness major 2, it might be helpful if the authors could provide further explanation or analysis on how the Gaussian-based representation affects generalization across different architectures.
3. It might be helpful if the authors could provide some discussion on the scalability of the proposed method to higher-resolution datasets, e.g., full size and resolution ImageNet-1K. Also, could the same Gaussian-based representation be extended to detection or segmentation tasks?

---

> ### Author Response · Authors · 2025-11-19
>
> We sincerely thank you for your recognition of our writing, the idea of GSDD, and its performance. Below we provide detailed responses and additional experiments regarding your questions on GPU dependency, generalization ability of Gaussian representations, and scalability to higher-resolution datasets.
>
> > **W1&Q1**: The efficiency of the proposed method relies heavily on CUDA-based parallel rasterizer, which raises concerns about portability and reproducibility.
>
> **Summary**: We benchmarked the operator on five distinct GPU architectures, and its memory usage and accuracy remained highly consistent, ensuring reproducibility and stability.
>
> We fully agree that GPU-specific operators may raise portability concerns.
> To evaluate the stability of our CUDA-based rasterizer, we conducted systematic benchmarking on **five different GPU architectures**. We used an image-fitting task in which Gaussian representations reconstruct **1000 images sampled from ImageNet** (10 classes × 100 images each). Each experiment was repeated three times, and we averaged all results. We tested four storage budgets (21, 170, 682, 2730 Gaussians per image). All experiments used **PyTorch 2.5.1** and **CUDA ≥ 12.0**.
>
> Table 1: Runtime (minutes)
>
> | #Points | Tesla-V100-SXM2-32GB | NVIDIA-A100-SXM4-40GB | NVIDIA-GeForce-RTX-3090 | NVIDIA-GeForce-RTX-4090 | NVIDIA-GeForce-RTX-5090 |
> | ------- | -------------------- | --------------------- | ----------------------- | ----------------------- | ----------------------- |
> | 21      | 23.65                | 10.45                 | 10.33                   | 7.60                    | 7.13                    |
> | 170     | 26.11                | 12.87                 | 16.07                   | 9.38                    | 6.99                    |
> | 682     | 30.28                | 19.09                 | 25.44                   | 14.06                   | 9.54                    |
> | 2730    | 42.94                | 41.53                 | 61.38                   | 31.00                   | 21.79                   |
>
> Different GPUs exhibit different runtimes due to heterogeneous compute capabilities, and 5090 is the fastest, while 3090 is the slowest.
>
> Table 2: Peak GPU memory (GB)
>
> | #Points | Tesla-V100-SXM2-32GB | NVIDIA-A100-SXM4-40GB | NVIDIA-GeForce-RTX-3090 | NVIDIA-GeForce-RTX-4090 | NVIDIA-GeForce-RTX-5090 | MAX-MIN |
> | ------- | -------------------- | --------------------- | ----------------------- | ----------------------- | ----------------------- | ------- |
> | 21      | 1.66                 | 1.66                  | 1.66                    | 1.66                    | 1.66                    | 0.00    |
> | 170     | 1.70                 | 1.70                  | 1.70                    | 1.70                    | 1.70                    | 0.00    |
> | 682     | 1.81                 | 1.81                  | 1.81                    | 1.81                    | 1.81                    | 0.00    |
> | 2730    | 2.28                 | 2.28                  | 2.28                    | 2.28                    | 2.28                    | 0.00    |
>
> Memory usage remained **identical across all platforms**.
>
> Table 3: Reconstruction Quality (PSNR)
>
> | #Points | Tesla-V100-SXM2-32GB | NVIDIA-A100-SXM4-40GB | NVIDIA-GeForce-RTX-3090 | NVIDIA-GeForce-RTX-4090 | NVIDIA-GeForce-RTX-5090 | MAX-MIN |
> | ------- | -------------------- | --------------------- | ----------------------- | ----------------------- | ----------------------- | ------- |
> | 21      | 20.24                | 20.24                 | 20.24                   | 20.24                   | 20.23                   | 0.01    |
> | 170     | 25.93                | 26.03                 | 26.03                   | 26.02                   | 26.02                   | 0.10    |
> | 682     | 30.65                | 31.06                 | 31.06                   | 31.03                   | 31.03                   | 0.41    |
> | 2730    | 36.09                | 36.08                 | 36.08                   | 36.06                   | 36.06                   | 0.03    |
>
> PSNR variation is extremely small across architectures.
> Slight performance drops on V100 are expected due to older hardware, but results on A100/3090/4090/5090 remain highly consistent.
>
> **Conclusion:** With modern PyTorch (2.5.1) and CUDA (≥ 12.0), our CUDA operator behaves consistently across major GPU families, demonstrating good portability and reproducibility.

---

> ### Author Response · Authors · 2025-11-19
>
> > **W2&Q2**: As shown in Tables 4 and 10, GSDD performance improvement across different architectures is relatively limited compared to performance on the same architecture. This paper only reports the average results for each architecture without providing detailed comparisons per architecture.
>
> We have updated Table 10 (Table 14 for latest manuscript) with a more detailed comparison with other methods across four architectures including AlexNet, VGG11, ResNet18 and ViT.
>
> | **Test Net** | **Method** | **Nette**      | **Woof**       | **Fruit**      | **Yellow**     | **Meow**       | **Squawk**     |
> | ------------ | ---------- | -------------- | -------------- | -------------- | -------------- | -------------- | -------------- |
> | **AlexNet**  | **TM**     | 13.2 ± 0.6     | 10.0 ± 0.0     | 10.0 ± 0.0     | 11.0 ± 0.2     | 9.8 ± 0.0      | -            |
> |              | **IDC**    | 17.4 ± 0.9     | 16.5 ± 0.7     | 17.9 ± 0.7     | 20.6 ± 0.9     | 16.8 ± 0.5     | 20.7 ± 1.0     |
> |              | **FreD**   | 35.7 ± 0.4     | 23.9 ± 0.7     | 15.8 ± 0.7     | 19.8 ± 1.2     | 14.4 ± 0.5     | 36.3 ± 0.3     |
> |              | **DDiF**   | 60.7 ± 2.3     | **36.4 ± 2.3** | 41.8 ± 0.6     | **56.2 ± 0.8** | **40.3 ± 1.9** | **60.5 ± 0.4** |
> |              | **GSDD**   | **63.1 ± 1.3** | 33.6 ± 0.9     | **41.9 ± 2.0** | 53.5 ± 0.9     | 38.1 ± 0.8     | 60.1 ± 0.2     |
> | ---          | ---        | ---            | ---            | ---            | ---            | ---            | ---            |
> | **VGG11**    | **TM**     | 17.4 ± 2.1     | 12.6 ± 1.8     | 11.8 ± 1.0     | 16.9 ± 1.1     | 13.8 ± 1.3     | -             |
> |              | **IDC**    | 19.6 ± 1.5     | 16.0 ± 2.1     | 13.8 ± 1.3     | 16.8 ± 3.5     | 13.1 ± 2.0     | 19.1 ± 1.2     |
> |              | **FreD**   | 21.8 ± 2.9     | 17.1 1.7       | 12.6 ± 2.6     | 18.2 ± 1.1     | 13.2 ± 1.9     | 18.6 ± 2.3     |
> |              | **DDiF**   | 53.6 ± 1.5     | 29.9 ± 1.9     | 33.8 ± 1.9     | 44.2 ± 1.7     | **32.0 ± 1.8** | 37.9 ± 1.5     |
> |              | **GSDD**   | **56.8 ± 1.6** | **32.7 ± 1.4** | **34.1 ± 3.0** | **57.3 ± 4.5** | 31.3 ± 1.8     | **55.9 ± 3.4** |
> | ---          | ---        | ---            | ---            | ---            | ---            | ---            | ---            |
> | **ResNet18** | **TM**     | 34.9 ± 2.3     | 20.7 ± 1.0     | 23.1 ± 1.5     | 43.4 ± 1.1     | 22.8 ± 2.2     | -              |
> |              | **IDC**    | 43.6 ± 1.3     | 23.2 ± 0.8     | 32.9 ± 2.8     | 44.2 ± 3.5     | 28.2 ± 0.5     | 47.8 ± 1.9     |
> |              | **FreD**   | 48.8 ± 1.8     | 28.4 ± 0.6     | 34.0 ± 1.9     | 49.3 ± 1.1     | 29.0 ± 1.8     | 50.2 ± 0.8     |
> |              | **DDiF**   | **63.8 ± 1.8** | 37.5 ± 1.9     | 42.0 ± 1.9     | **55.9 ± 1.0** | 35.8 ± 1.8     | **62.6 ± 1.5** |
> |              | **GSDD**   | 59.1 ± 3.0     | **39.5 ± 2.0** | **42.6 ± 0.8** | **55.9 ± 1.2** | **40.9 ± 2.9** | **62.6 ± 0.6** |
> | ---          | ---        | ---            | ---            | ---            | ---            | ---            | ---            |
> | **ViT**      | **TM**     | 22.6 ± 1.1     | 15.9 ± 0.4     | 23.3 ± 0.4     | 18.1 ± 1.3     | 18.6 ± 0.9     | -              |
> |              | **IDC**    | 31.0 ± 0.6     | 22.4 ± 0.8     | 31.1 ± 0.8     | 30.3 ± 0.6     | 21.4 ± 0.7     | 32.2 ± 1.2     |
> |              | **FreD**   | 38.4 ± 0.7     | 25.4 ± 1.7     | 31.9 ± 1.4     | 37.6 ± 2.0     | 19.7 ± 0.8     | 44.4 ± 1.0     |
> |              | **DDiF**   | **59.0 ± 0.4** | **32.8 ± 0.8** | 39.4 ± 0.8     | **47.9 ± 0.6** | **27.0 ± 0.6** | **54.8 ± 1.1** |
> |              | **GSDD**   | 53.4 ± 0.6     | 32.6 ± 1.1     | **41.0 ± 1.2** | 47.6 ± 0.8     | 25.8 ± 0.4     | 53.5 ± 2.1     |
>
> Overall, GSDD achieves competitive performance compared to DDiF (but better on distillation architecture and faster), and outperforms the other methods.

---

> ### Author Response · Authors · 2025-11-19
>
> > **W3&Q3**: While the proposed method demonstrates efficiency and scalability, experiments have primarily focused on relatively low-resolution datasets, such as subsets of CIFAR and ImageNet (low-resolution version).
>
> We agree that evaluating on larger datasets and higher resolutions is crucial.
> We first report results on **ImageNet-Subset at 256 resolution**:
>
> |          | Nette    | Woof     | Fruit    | Yellow   | Meow     | Squawk   |
> | -------- | -------- | -------- | -------- | -------- | -------- | -------- |
> | Vanilla  | 32.1     | 20.0     | 19.5     | 33.4     | 21.2     | 27.6     |
> | IDC      | 53.7     | 30.2     | 33.1     | 52.2     | 34.6     | 47.0     |
> | FreD     | 54.2     | 31.2     | 32.5     | 49.1     | 34.0     | 43.1     |
> | LatentDD | 56.1     | 28.0     | 30.7     | —        | 36.3     | 47.1     |
> | DDiF     | 67.8     | 39.6     | 43.2     | 63.1     | 44.8     | 67.0     |
> | GSDD     | **70.0** | **42.6** | **51.2** | **67.4** | **46.4** | **70.4** |
>
> GSDD achieves the best performance on all subsets.
>
> We further apply GSDD to **ImageNet-1K**, using SRe2L as the baseline. We additionally evaluate SRe2L with IDC, where IDC downsamples each image by a factor of k, thereby increasing the total number of images by k times.
>
> |  | IPC=1| IPC=10|
> | - | - | - |
> | SRe2L|0.9| 21.3|
> | SRe2L+IDC | 13.4(k=20)|41.5(k=4)|
> | SRe2L+GSDD | 29.0(GPC=50)| 46.2(GPC=200)|
>
> Best results for IDC are selected from $k \in \lbrace 4,20,50\rbrace$ for IPC=1, and $k\in \lbrace 4,10,20\rbrace$ for IPC=10.
>
> It can be seen that our method boosts the performance of SRe2L, especially for the IPC=1 setting, improving its accuracy from 0.9 to 29.0. In addition, our method can further enhance the IPC=10 performance from 21.3 to 42.2. It is worth noting that this is the first time a dataset-parameterization–based method in DD has ever been applied to the ImageNet-1K dataset, thanks to efficient distillation framework like SRe2L, as well as the efficiency and representativeness of GSDD.
>
> > **Q4**: Could the same Gaussian-based representation be extended to detection or segmentation tasks?
>
> Expanding beyond image classification is a central topic in dataset distillation research. Recent pioneering works have extended DD to **object detection** and **action segmentation** [1–3].
> In principle, our explicit Gaussian representation has the potential to be extended to detection and segmentation from two perspectives:
>
> 1. **Pixel-level optimization**
>    Dataset distillation optimizes pixel-wise representations; hence any differentiable representation (including Gaussian splatting) is theoretically compatible with detection/segmentation.
> 2. **Advantages for structured tasks**
>    - GSDD uses explicit, decomposable 2D Gaussians that may allow **foreground-background decomposition**.
>    - Gaussian splatting has already been widely used for **semantic and 3D scene understanding** [4–5], indicating its potential for pixel-structured tasks.
>
> Most of the results have already been updated in the manuscript, including results on different GPU architecture, more detailed cross-architecture comparison, and results on higher resolution.
>
> Thank you again for your time and valuable feedback, and please feel free to let us know if you have any further questions.
>
> References:
>
> [1] Fetch and forge: Efficient dataset condensation for object detection, NeurIPS, 2024.
>
> [2] OD3: Optimization-free dataset distillation for object detection, 2025.
>
> [3] Condensing Action Segmentation Datasets via Generative Network Inversion, CVPR, 2025
>
> [4] Language embedded 3d gaussians for open-vocabulary scene understanding, CVPR, 2024
>
> [5] Segment any 3d gaussians, AAAI, 2025

---

### Official Review · Reviewer_45iu · 2025-10-31

**Soundness:** 3
**Presentation:** 3
**Contribution:** 3
**Rating:** 4
**Confidence:** 5

**Summary:**

This study points out that existing dataset distillation methods using dense representations, which is hard to capture the relative importance of pixels effectively, leading to redundancy. To achieve efficient sparse representations, this study proposes Gaussian splatting dataset distillation (GSDD), which utilizes Gaussian splatting to parameterize the distilled synthetic images. Through the experimental results, GSDD shows the efficiency and effectiveness improvement.

**Strengths:**

1. To my knowledge, this is the first work that employs Gaussian splatting in the dataset distillation. Gaussian splatting is widely researched and utilized across diverse fields, thus possessing high potential for extension.

2. The detailed analysis of the Gaussian representation described in Section 3.3 is highly beneficial as it aids in a more thorough understanding of the GSDD. In particular, the significant variation in performance depending on large opacity and size is of particular interest.

3. GSDD achieves strong performance with significant gap on some benchmark datasets.

**Weaknesses:**

1. There is a lack of clear justification for the motivation to apply Gaussian splatting to dataset distillation. I believe the current manuscript focuses more on methodologies for efficiently using Gaussian splatting in Dataset distillation than on unique characteristics of Gaussian splatting compared to existing parameterization methods. The issues of redundancy and computational overhead highlighted by the authors as problems in prior research have already been noted in several previous studies[1,2,3,4]. Specifically, while the paper highlights efficiency concerns regarding time and memory for DDiF[5] (the core baseline adopted in this study), this does not sufficiently justify shifting the framework to Gaussian splatting without improving INR-based parameterization. Therefore, an in-depth analysis is required to present new limitations of existing prior research and demonstrate how Gaussian splatting's inherent characteristics can address these.

2. There is insufficient evidence to support the claim that GSDD achieves higher performance than prior research. The contributions of this study are believed to be the first application of Gaussian splatting in the field of dataset distillation and the enhancement of efficiency through various techniques. While this approach achieves high efficiency in terms of time and memory, the lack of evidence and explanation as to why it outperforms prior research makes it difficult to accept. For instance, DDiF argued that the introduction of INR yields higher expressiveness despite using a smaller budget, through theoretical analysis and reconstruction experiments. Furthermore, experiments with a fixed number of decoded instances supported the notion that DDiF's high performance stems from its high expressiveness and diversity. Similarly, a deeper analysis is required to understand the mechanism through which the proposed idea, GSDD, achieves its high performance.

3. There is no analysis of whether GSDD can be applied across diverse application scenarios and achieve high performance. Parameterization methods effective only in specific application area have limitations in their applicability. Several prior studies report experimental results on corrupted datasets[2,3,5] and cross-resolution[5] to evaluate the out-of-domain generalization of each methods. Furthermore, as the primary baseline DDiF is a unified framework for grid-based data across various modalities, it provided performance comparison results across image, video, audio, and 3D voxel datasets. Demonstrating GSDD's superior performance across diverse application scenarios would significantly enhance the impact and applicability of this research.

[1] A Comprehensive Survey of Dataset Distillation

[2] Frequency Domain-based Dataset Distillation

[3] Sparse Parameterization for Epitomic Dataset Distillation

[4] Generalizing Dataset Distillation via Deep Generative Prior

[5] Distilling Dataset into Neural Field

**Questions:**

1. I am curious about the performance of Gaussian splatting-based parameterization without applying the various techniques for improving efficiency (Section 3.2). I am also curious about the performance improvement when adding each technique, extending Table 3.

2. I am also curious whether GSDD also demonstrates high performance and efficiency for image datasets with resolutions greater than 128.

3. Recently, research into soft labels has also been actively explored as an alternative to one-hot labels[6,7,8]. While this work defined by one-hot label encoding (Line 98), I would be interested to see the performance of GSDD when soft label is applied

4. Quantization using bf16 directly influences the final image per class calculation (Line 402–408), so it affects diversity and playing a key role in GSDD performance (Table 3). However, this technique is not limited to GSDD and can also be applied to existing parameterization methods. Therefore, I would like to know the performance comparison when quantization is applied to existing parameterization methods.

5. I understood that each Gaussian primitive contributes to a single distilled image. I am curious whether this could be extended to allow a Gaussian primitive to contribute to multiple distilled images. If possible, this could further enhance efficiency.

[6] A Label is Worth A Thousand Images in Dataset Distillation

[7] GIFT: Unlocking Full Potential of Labels in Distilled Dataset at Near-zero Cost

[8] Heavy Labels Out! Dataset Distillation with Label Space Lightening

---

> ### Author Response · Authors · 2025-11-19
>
> We sincerely appreciate your support regarding the prospects, experimental analysis, and performance of our work. And thanks for your rigorous requirements regarding our motivation and valuable suggestions on more extensions.
>
> > **W1:** There is a lack of clear justification for the motivation to apply Gaussian splatting to dataset distillation. An in-depth analysis is required to present new limitations of existing prior research and demonstrate how Gaussian splatting's inherent characteristics can address these.
>
> Existing parameterized distillation works can be categorized into **Inter-image Redundancy Removal** (e.g., [1-4], using bases/decoders to store shared knowledge) and **Intra-image Redundancy Removal** (e.g., [5,6], using frequency domains or INRs to reduce single-image parameters).
>
> **Limitations of existing methods**:
>
> * Complex Network Design (Inter-image): Methods like SPEED [3] and HaBa [2] require complex network structures to extract shared knowledge.
>
>   - *Complex Storage Estimation:* For example, SPEED's storage formula is $DK+1.5NHk+R(3D^2+7D)+L(D+1)$,where every symbol is a hyperparameter.
>
>   - *Design Difficulty:* It is difficult to determine the optimal structure for a given budget and resolution, e.g. solve SPEED hypers $D,K,k,R,\cdots$ given a storage budget. Even sometimes no valid solution exists for low budgets at high resolutions (e.g. HaBa).
>
> * Efficiency & Optimization Limits (Intra-image): FreD [5] optimizes basis parameters but not the basis itself. DDiF [6], while more flexible, is limited in efficiency due to point-wise querying.
>
> **Advantages of GSDD**:
>
> - **Structural Simplicity:** We only need to set GPC (Gaussian Images Per Class).
> - **Efficiency & High Quality**: GSDD is able to balance both representational quality and efficiency.
>
> We benchmark different parameterization methods using the ImageNette dataset with same storage, by fitting the first 100 images from each class (1,000 images in total) at a resolution of 128. Optimization steps is set to 1,000, and for each algorithm we search for the optimal lr and optimizer.
>
> Table 1: Benchmark of representativeness and efficiency
>
> |  | FreD | SPEED | DDiF | GSDD  |
> | - | - | - | - | - |
> | PSNR↑ | 15.69 | 17.77 | **23.23** | 23.22  |
> | GPU Memory(MB)↓ | 280.4  | 298.5 | 1078.8  | **150.2** |
> | Time(s)↓  | **15.5** | 69.4  | 1088.4 | 24.5 |
>
> **Compared with existing methods, GSDD achieves excellent performance in terms of representational capacity, GPU memory usage, and time cost.**
>
> Moreover, we would like to clarify that our intention is to offer an alternative, rather than *“shifting the framework to Gaussian Splatting without improving INR-based methods.”* DDiF exhibits strong advantages in cross-modal and cross-resolution tasks, and we acknowledge that DDiF still holds substantial potential for improvement, for example by using one single DDiF to represent multiple images.
>
> In addition, we believe that explicit and implicit representations can be further combined to encode structural information and feature jointly.
>
> > **W2:** There is insufficient evidence to support the claim that GSDD achieves higher performance than prior research. The lack of evidence and explanation as to why it outperforms prior research makes it difficult to accept.
>
> Diversity is crucial for DD[7-10]. However, existing methods struggle to maximize diversity due to the trade-off between representation quality and efficiency. In our response to **W1**, we show GSDD’s representational efficiency. We now present experiments demonstrating that GSDD’s efficiency enables further diversity, leading to improved performance.
>
> Specifically, we evaluate the effect of **GPC (Gaussian Images Per Class)** under various datasets, distillation algorithms(TM[11] and DM[12]), and storage budgets:
>
> Table 2: CIFAR10，IPC=1, with TM
>
> |GPC|20|30|40|50|60 |80|100|
> |-|-|-|-|-|-|-|-|
> | DD_acc | 65.5 | 66.9 | **67.5** | 64.6 | 67.1 |63.6|58.1|
>
> Table 3: CIFAR10，IPC=10, with TM
>
> | GPC | 1 | 10  | 40 | 80 | 160|
> | - | - | - | - | - | - |
> | DD_acc | 41.4 | 62.0 | 68.9 | 74.7 | **76.0** |
>
> Table 4: ImageNette, IPC=1, with TM
>
> | GPC | 4 | 16 | 64| 128  |
> | - | - | - | - | - |
> | DD_acc | 63.2 | 71.0  | 76.4 | **76.6** |
>
> Table 5: ImageNette, IPC=1, with DM
>
> | GPC | 100 |120 | 160|200|240 |
> |-|-|-|-|-|-|
> | DD_acc|72.2|74.4|72.2|**74.6**|71.0|
>
> We observe that DD performance generally increases with larger GPC. Performance only begins to drop when the storage budget is extremely small and GPC becomes so large that the representational capacity is significantly weakened. GSDD benefits from highly efficient representations, allowing GPC to be expanded. This enables us to directly assess how increasing diversity impacts distilled dataset performance.
>
> This finding offers a interesting perspective, one can trade off per-image representation quality by adopting very sparse representations in order to dramatically increase overall diversity and thereby improve accuracy.

---

> ### Author Response · Authors · 2025-11-19
>
> > **W3**: There is no analysis of whether GSDD can be applied across diverse application scenarios and achieve high performance.
>
> One of the key advantages of GSDD over traditional parameterized approaches is its inherent ability to perform **cross-resolution distillation**. Gaussian Splatting naturally supports rendering at arbitrary resolutions through supersampling, enabling us to distill data at a low resolution (e.g., 128) while still using it to train models at a higher resolution (e.g., 256). This effectively reduces the computation when distilling datasets at large resolutions. Following results in DDiF, we conduct cross-resolution experiments as shown below:
>
> Table6: Cross-resolution results, distilled with low resolution(128) and test at higher resolution(256/512). $difference=Acc_{128}-Acc_{256/512}$, $ratio = difference/Acc_{256}$
>
> | test resolution | test network | method  | accuracy↑ | difference↓ | ratio↓   |
> | - | - | - | - | - | - |
> | 256| ConvNetD5    | Vanilla | 31.2| 20.2| 0.39|
> |  |  | IDC| 55.0| 6.4| 0.1|
> |  | | SPEED   | 58.8| 8.1| 0.12|
> |  |  | FreD| 56.4| 10.4| 0.16|
> | |  | DDiF| 66.3| **5.7**| **0.08** |
> |  |   | GSDD| **66.4**  | 10.0| 0.13     |
> | | ConvNetD6    | Vanilla | 44.0| 7.3| 0.14     |
> |  | | IDC| 55.4| 6.0| 0.10|
> |  | | SPEED   | 62.6| 4.3| 0.06|
> |  | | FreD| 61.8| 5.0| 0.07|
> | | | DDiF| 70.6| **1.4**| **0.02** |
> |  | | GSDD| **71.6**| 4.8| 0.06     |
> | 512| ConvNetD5| Vanilla | 27.4| 24.3| 0.47|
> | | | IDC| 39.5| 21.9| 0.36|
> | | | SPEED   | 45.0| 21.9| 0.33|
> | | | FreD| 42.9| 23.9| 0.36|
> | | | DDiF| 58.7| **13.3**| **0.18** |
> | | | GSDD| **59.4**  | 17.0| 0.22|
> | | ConvNetD6| Vanilla | 41.2| 10.1| 0.20|
> | | | IDC| 51.5| 9.9| 0.16|
> | | | SPEED   | 60.1| 6.8| 0.10|
> | | | FreD| 56.3| 10.5| 0.16|
> | | | DDiF| **69.0**  | **3.0**| **0.04** |
> | | | GSDD| 67.8| 8.6| 0.11|
>
> For IDC, SPEED, and FreD, we adopt their best-performing upsampling strategies. For DDiF and GSDD, supersampling is applied, allowing lossless upscaling of the distilled images.
>
> GSDD maintains competitive performance in cross-resolution tasks. Although its relative performance drop is larger than DDiF, which is likely due to GSDD’s inherent sparsity, reducing its ability to capture fine-grained features when evaluated at higher resolutions.
>
> Most existing DD methods focus exclusively on the image modality, and extension to other modalities has recently been an active area of research. Dataset parameterization methods, in particular, are typically constrained to a single modality because their data structures are designed specifically for images [1–5]. As a pioneering work, DDiF [6] provided a unified representation across modalities. GSDD is currently developed for the image modality, it holds promise for extensions to 3D and video dataset distillation, provided that the corresponding CUDA operators are implemented.
>
> > **Q1**: The performance of Gaussian splatting-based parameterization without applying the various techniques for improving efficiency, and the performance improvement when adding each technique, extending Table 3.
>
> We performed an incremental experiment starting from a baseline with no techniques, and with each technique separately:
>
> Table 7: extensive ablation study
>
> | Setting | ImageNette Acc |
> | - | - |
> | baseline(none) | 74.2 |
> | w/ boundary    | 74.6 |
> | w/ bf16 | 76.0 |
> | w/ opacity | 76.0 |
> | w/ antilias | 74.4 |
>
> We observe that bf16 and opacity contribute most to the final performance, while boundary and antialias offer only minor gains. The slight inconsistencies with Table 3 likely arise because the effects of individual factors are hard to isolate, and their contributions do not necessarily combine linearly.
>
> However, our experiments consistently indicate that these factors generally contribute positively to performance, and their aggregated use leads to the strongest overall results.
>
> > **Q2**: Whether GSDD also demonstrates high performance and efficiency for image datasets with resolutions greater than 128.
>
>
> Table 8: Results on ImageNet Subsets at resolution of 256
>
> |          | Nette | Woof| Fruit    | Yellow   | Meow     | Squawk   |
> | -- | - | - | -- | - | -- | - |
> | Vanilla  | 32.1     | 20.0     | 19.5     | 33.4     | 21.2     | 27.6     |
> | IDC      | 53.7     | 30.2     | 33.1     | 52.2     | 34.6     | 47.0     |
> | FreD     | 54.2     | 31.2     | 32.5     | 49.1     | 34.0     | 43.1     |
> | LatentDD | 56.1     | 28.0     | 30.7     | —        | 36.3     | 47.1     |
> | DDiF     | 67.8     | 39.6     | 43.2     | 63.1     | 44.8     | 67.0     |
> | GSDD     | **70.0** | **42.6** | **51.2** | **67.4** | **46.4** | **70.4** |
>
> GSDD achieves the best performance across all datasets. We evaluate GSDD using the DM algorithm at 256 resolution with GPC = 120, and its improved diversity enables the DM algorithm to better capture the distributional differences between the distilled and real datasets.

---

> > ### Author Response · Authors · 2025-11-19
> >
> > > **Q3:** While this work defined by one-hot label encoding (Line 98), I would be interested to see the performance of GSDD when soft label is applied.
> >
> > Table 9: Results with soft-label
> >
> > | Loss Type | Nette| Woof| Fruit| Yellow| Meow| Squawk|
> > | - | - | - | - | - | - | - |
> > | onehot| 76.4| 46.4| **51.2** | **72.4** | **53.8** | **73.2** |
> > | kl ($\tau=3$)| 75.6| 46.0| 51.0| 68.6| 53.6| **73.2** |
> > | mixup+kl ($\tau$=3) | **77.2** | 45.8| 50.0| 70.2| 52.4| 72.2|
> > | mse| 75.4| 46.2| 50.0| 68.6| 49.6| 72.8|
> > | mixup+mse| 77.0| **49.4** | 48.2| 71.2| 52.0| 74.0|
> >
> > We first searched for the optimal hyperparameters on ImageNette and then applied them to the other subsets.
> > Soft labels are obtained by averaging the teacher-model predictions generated by TM during the buffer stage.
> > Importantly, we compute soft labels **only once** and reuse them throughout training, rather than repeatedly predicting soft labels across many epochs under different augmentation strategies.
> >
> > The experiments show that soft labels do not yield appealing improvements. We hypothesize that this is because the distilled dataset produced by GSDD already possesses rich diversity, which may surpass the additional information provided by soft labels.
> >
> > > **Q4:**  Quantization technique is not limited to GSDD and can also be applied to existing parameterization methods. Therefore, I would like to know the performance comparison when quantization is applied to existing parameterization methods.
> >
> > We agree that quantization plays an important role. However, applying quantization to prior parameterization methods is non-trivial, different network architectures have different sensitivities to numerical precision and require carefully tuned optimizers. This involves substantial engineering effort and cannot be completed within the rebuttal period.
> >
> > Therefore, we remove quantization and compare all methods under fp32 for fairness.
> >
> > | imagenette | TM | FRePo | IDC  | FreD | HaBa |SPEED | NSD |GLaD | H-GLaD|DDiF| GSDD(w/o bf16) |
> > | - | - | - | - | - | - | - | - | - | - | - | - |
> > | acc | 51.4| 48.1|61.4|66.8|51.9|66.9|68.6|38.7|45.4|72.0|74.8|
> >
> > These results show that even without bf16, GSDD still achieves state-of-the-art performance.
> >
> > We apply quantization in GSDD for a deeper reason: traditional neural-network-based parameterization methods[1-4] tend to suffer from numerical instability and error accumulation after quantization. **In contrast, GSDD uses a differentiable rasterizer with explicit Gaussian primitives, directly projecting positions and colors, making it inherently robust to precision changes.**
> >
> > > **Q5:** Whether GSDD could be extended to allow a Gaussian primitive to contribute to multiple distilled images. If possible, this could further enhance efficiency.
> >
> > This is an excellent suggestion, and coincidentally, we explored this idea early in our project. Allowing a single Gaussian to represent multiple distilled images is appealing for efficiency, but we found it non-trivial due to two main challenges:
> >
> > 1. **Initialization difficulty.** It's hard to intialize GSDD with real images when Gaussian Images share common gaussians points.
> > 2. **Difficulty capturing shared image information.** Explicit Gaussian primitives struggle to extract shared latent representations across images, unlike implicit (network-based) methods.
> >
> > In summary, because GSDD uses **explicit representations**, it is much more difficult to reuse Gaussian primitives across images in the same way as neural-network-based methods[1-4]. Naively reusing Gaussians degenerates into low-level image augmentation rather than learning meaningful shared semantics.
> >
> > Unfortunately, our early code and the final GSDD implementation are incompatible due to operator-level changes and redesigned class structures, so it's hard to provide results during the rebuttal. However, we hope our experience helps clarify this question.
> >
> > Once again, we sincerely appreciate your interest in our work and your thoughtful suggestions. Please feel free to let us know if you have any further questions.
> >
> > [1] Remember the Past: Distilling Datasets into Addressable Memories for Neural Networks, NeurIPS, 2022
> >
> > [2] Dataset Distillation via Factorization, NeurIPS, 2022
> >
> > [3] Sparse Parameterization for Epitomic Dataset Distillation, NeurIPS, 2023
> >
> > [4] Generalizing Dataset Distillation via Deep Generative Prior, CVPR, 2023
> >
> > [5] Frequency Domain-Based Dataset Distillation, NeurIPS, 2023
> >
> > [6] Distilling dataset into neural field, ICLR, 2025
> >
> > [7] On the Diversity and Realism of Distilled Dataset: An Efficient Dataset Distillation Paradigm, CVPR, 2024
> >
> > [8] Efficient Dataset Distillation via Minimax Diffusion, CVPR, 2204
> >
> > [9] Diversity-enhanced distribution alignment for dataset distillation, ICCV, 2025
> >
> > [10] DELT: a simple diversity-driven EarlyLate training for dataset distillation, CVPR, 2025
> >
> > [11] Dataset distillation by matching training trajectories, CVPR, 2022
> >
> > [12] Dataset Condensation With Distribution Matching, WACV, 2023

---

### Official Review · Reviewer_sSwA · 2025-10-31

**Soundness:** 2
**Presentation:** 3
**Contribution:** 2
**Rating:** 4
**Confidence:** 4

**Summary:**

This paper tackles the problem of data parameterization in dataset distillation (DD). It argues, correctly, that conventional dense pixel-grid representations are redundant, inefficient, and scale poorly. The authors propose GSDD (Gaussian Splatting Dataset Distillation), a novel and efficient parameterization that represents each distilled image as a sparse set of 2D Gaussian primitives. Each Gaussian is defined by 9 parameters (position, shape, color, opacity). These primitives are rendered into images using a highly parallelized, differentiable CUDA-based rasterizer. The central claims are that this sparse representation (1) is more storage-efficient, allowing for greater dataset diversity (more images per class) under a fixed budget , (2) enables a smoother optimization landscape for faster convergence , and (3) is computationally efficient. The method is evaluated on CIFAR-10, CIFAR-100, and ImageNet subsets , where it is shown to achieve state-of-the-art performance and significantly outperform the INR-based DDiF in speed and memory usage.

**Strengths:**

1. Novel and Clever Parameterization: The core idea of using a sparse set of 2D Gaussians to represent distilled images is highly original in the DD context. It elegantly sidesteps the high-frequency noise issues of pixel optimization (Fig 4) and the computational bottleneck of INR's per-pixel querying.

2. Efficiency vs. INRs: The paper provides compelling quantitative evidence (Figure 5) that GSDD is dramatically faster (both forward and backward) and more memory-efficient than the state-of-the-art functional parameterization, DDiF . This is a strong practical advantage.

3. Performance (on tested benchmarks): The method achieves state-of-the-art results on all tested benchmarks (CIFAR-10, CIFAR-100, and ImageNet subsets), outperforming numerous baselines across various IPC settings.

**Weaknesses:**

1. Critical Omission of ImageNet-1K: The most significant weakness is the failure to evaluate on full ImageNet-1K. The paper is titled "Efficient Dataset Distillation" and repeatedly claims scalability. However, it avoids the standard large-scale benchmark where scalability is truly tested. Recent SOTA methods in scalable DD (e.g., SRe2L, RDED, etc.) all report 1K results. Without this comparison, the central claim of scalability is unsubstantiated. The results on "ImageNet subsets"  are insufficient.

2. Confounding Initialization: The initialization procedure  is a major methodological flaw. The model is "pre-trained" to match real images via MSE loss. This raises two problems:
   - Performance Confound: How much of the final SOTA performance is simply due to this high-fidelity, real-data-based "warm start" rather than the superiority of the GSDD parameterization within the distillation process?
   - Privacy Contradiction: This initialization method directly leaks data from the original dataset, which contradicts the paper's (and the field's) stated motivations for privacy preservation. A method that requires fitting to real images cannot be straightforwardly used in privacy-sensitive scenarios.

3. Hyperparameter Complexity: The method introduces a critical new hyperparameter trade-off: the number of Gaussians per image ($M$) vs. the number of Gaussian Images Per Class (GPC). While the paper explores this trade-off (Fig 3c, 3d), it's unclear how to optimally set these values for a new dataset and budget, making the method less of a simple drop-in replacement than advertised.

**Questions:**

1. ImageNet-1K: The most pressing question is the lack of full ImageNet-1K experiments. Given the paper's focus on efficiency and scalability, why was this benchmark omitted? Can you provide results for GSDD on ImageNet-1K, comparing it to scalable SOTA methods like SRe2L and RDED?

2. Initialization Ablation: What is the performance of GSDD if the Gaussians are initialized randomly (e.g., random positions/colors, small isotropic covariances) instead of being pre-fit to real images? This is a crucial ablation to understand the true source of the performance gains and to validate the method's feasibility for privacy-preserving applications.

3. Privacy: How do you reconcile the claim of supporting "privacy-preserving research" with an initialization method  that explicitly fits the synthetic data to real data samples?

4. Gaussian Escape: In your ablation (Table 3), the "w/o boundary" setting shows a performance drop. Is this drop due to removing the loss term (Eq 10) or the hard clipping (Eq 11)? What happens if you only use the loss, or only the clipping?

I am willing to raise my score if the authors can provide a comprehensive comparison on ImageNet-1K against relevant SOTA methods (like SRe2L and RDED) and address the confounding effect of their real-image-based initialization in their rebuttal.

---

> ### Author Response · Authors · 2025-11-19
>
> We sincerely thank the reviewer for recognizing the parameterization, efficiency, and performance of our Gaussian representation. In response to your questions, we report the results on the large-scale ImageNet-1K dataset and the ImageNet-Subset (256 resolution). We also provide results on different initialization strategies, the impact of hyperparameters (and how to determine the optimal GPC), and a fine-grained ablation study on Gaussian Escape.
>
> >  **W1&Q1:** Critical Omission of ImageNet-1K: The most significant weakness is the failure to evaluate on full ImageNet-1K.
>
> **Summary:** To the best of our knowledge, there has been no prior exploration of distilled dataset parameterization techniques on the ImageNet dataset. **We provide the first application of dataset parameterization to ImageNet-1K**, and we additionally report results on the ImageNet-Subset (256×256).
>
> Table 1: ImageNet-1K Results (based on SRe2L [12])
>
> |  | IPC=1| IPC=10|
> |-|-|-|
> |SRe2L|0.9| 21.3|
> |SRe2L+IDC|13.4(k=20)|41.5(k=4)|
> |SRe2L+GSDD|29.0(GPC=50)|46.2(GPC=200)|
>
> Best results for IDC are selected from $k \in \lbrace 4,20,50\rbrace$ for IPC=1, and $k\in \lbrace 4,10,20\rbrace$ for IPC=10.
>
> Since the RDED algorithm does not perform optimization on images, we only evaluate SRe2L, which inspired subsequent works such as EDC, GVBSM, et al. The results show that our method substantially boosts the performance of SRe2L, especially in the IPC=1 setting. Here, $k$ denotes the resizing ratio used in IDC.
>
> In addition, we supplement results on the higher-resolution (256×256) ImageNet Subset.
>
> Table 2: ImageNet Subsets (256×256) (based on DM [11])
>
> | |Nette|Woof|Fruit|Yellow|Meow|Squawk|
> |-|-|-|-|-|-|-|
> |Vanilla|32.1|20.0|19.5|33.4|21.2|27.6|
> |IDC|53.7|30.2|33.1|52.2|34.6|47.0|
> |FreD|54.2|31.2|32.5|49.1|34.0|43.|
> |LatentDD|56.1|28.0|30.7 | - | 36.3  | 47.1 |
> | DDiF | 67.8 | 39.6 | 43.2 | 63.1 | 44.8 | 67.0 |
> | GSDD | **70.0** | **42.6** | **51.2** | **67.4** | **46.4** | **70.4** |
>
> Table 2 shows that our method achieves SOTA performance on higher-resolution datasets.
>
> >  **W2&Q2&Q3**: The initialization procedure raises two problems including **performance confound** and **privacy contradiction**.
>
> We fully understand your concerns. Since both Q2 and Q3 relate to the initialization procedure, we will explain them together.
>
> **Summary:** Real image initialization is a standard technique in DD [1-6]. And at large compression ratios, the privacy risk of this technique is comparable to random initialization (difference <1%). Notably, our method achieves excellent performance (69.4) even **without** real image initialization.
>
> **Using real images to init is a critical technique for performance enhancement and is widely considered a default part of the DD pipeline [1-6].** The primary goal of DD is data efficiency; privacy is a derivative application, and standard DD literature typically does not benchmark privacy specifically.
>
> **Regarding potential privacy leakage of DD**, [7] provides a pioneering exploration in Section IV.C:
>
> Table 3: The AUC of Membership Inference Attacks (MIA) on a subset of CIFAR10
>
> | compression ratio | Init   | DSA  | DM | IDC | RFAD |
> | - | - | - | - | - | - |
> | 0.25 | random | 73.67 | 75.08  | 86.47 | 65.84 |
> | | real   | **76.46** | **77.73** | **98.07** | **74.86** |
> | 0.125 | random | **63.40** | **63.64** | 69.86     | 58.00 |
> |  | real   | 61.89 | 63.23 | **71.59** | **68.33** |
> | 0.05| random | **55.06** | 55.15  | 59.09  | 56.31 |
> |  | real   | 55.03  | **55.59** | **59.73** | **67.00** |
> | 0.025 | random | 52.38  | 52.35 | 54.62 | 54.37 |
> | | real   | **52.52** | **52.89** | **56.34**| **63.67** |
>
> *Note: Higher AUC indicates greater privacy leakage (50 = random guessing).* As compression increases (ratio decreases), attack effectiveness weakens. **Although real initialization yields slightly higher AUC than random initialization, the difference is generally marginal.** We also provide visualizations in the Appendix showing that distilled images differ significantly from the originals.
>
> **Ablation on Initialization:** We found that optimizing Gaussian representations from random initialization is unstable. Random Gaussian points are initially too small and sparse (large gaps), preventing effective reception of gradient signals.
>
> To address this, we utilize a Solid Color Warmup. All Gaussian representations first fit a single solid color image to reduce gaps and adjust point size, followed by distillation. **This process does not fit any real images.**
>
> Table 4: Performance Comparison of Initialization Strategies
>
> | | IDC(w/ init) | FreD(w/ init) | NSD(w/o init) | DDiF(w/ init) | GSDD(w/o init) | GSDD(w/o init, warmup) | GSDD(w/ init) |
> | - | - | - | - | - | - | - | - |
> | ImageNette | 61.4  | 66.8  | 68.6 | 72.0  | 63.6  | 69.4 | 76.4 |
>
> **Even without real image initialization, our method (w/ warmup) achieves excellent results**, outperforming several methods that rely on real image initialization.

---

> ### Author Response · Authors · 2025-11-19
>
> > **W3:** Hyperparameter Complexity: The method introduces a critical new hyperparameter trade-off: the number of Gaussians per image vs. the number of Gaussian Images Per Class (GPC). how to optimally set these values for a new dataset and budget?
>
> **Summary:** We conducted extensive ablations on the Gaussian Images Per Class (GPC). Generally, a larger GPC yields better results. We discovered a positive correlation between the performance of the "Initialization Dataset" (sampled real images and fit by GSDD as initialization, iter=0) and the final distilled performance. This allows us to estimate the optimal GPC using only the initialization dataset. **Compared to existing methods, GSDD is more hyperparameter-friendly.**
>
> `init_acc`: Performance of initialized gaussian images (initialized on sampled real images). `DD_acc`: Performance after distillation.
>
> Table 5: CIFAR10, IPC=1, TM[10]
>
> | GPC      | 20   | 30   | 40       | 50   | 60   | 80 | 100|
> | -------- | ---- | ---- | -------- | ---- | ---- | - | - |
> | init_acc | 39.3 | 40.3 | **43.1** | 41.8 | 41.6 | 39.6 | 35.2|
> | DD_acc   | 65.5 | 66.9 | **67.5** | 64.6 | 67.1 |63.6 | 58.1 |
>
> Table 6: CIFAR10, IPC=10, TM[10]
>
> | GPC      | 1    | 10   | 40   | 80   | 160      |
> | -------- | ---- | ---- | ---- | ---- | -------- |
> | init_acc | 15.3 | 32.9 | 49.4 | 56.8 | **61.1** |
> | DD_acc   | 41.4 | 62.0 | 68.9 | 74.7 | **76.0** |
>
> Table 7: ImageNette, IPC=1, TM[10]
>
> | GPC      | 4    | 16   | 64   | 128      |
> | -------- | ---- | ---- | ---- | -------- |
> | init_acc | 28.4 | 71   | 58   | **61.4** |
> | DD_acc   | 63.2 | 71   | 76.4 | **76.6** |
>
> Table 8: ImageNette, IPC=1, DM[11]
>
> | GPC      | 100  | 120  | 160  | 200      | 240  |
> | -------- | ---- | ---- | ---- | -------- | ---- |
> | init_acc | 61.6 | 62.2 | 62.8 | **65.6** | 64.8 |
> | DD_acc   | 72.2 | 74.4 | 72.2 | **74.6** | 71.0 |
>
> The optimal GPC range for `init_acc` and `DD_acc` is consistent. This is intuitive: increasing GPC reduces the number of Gaussian points per image under fixed storage, creating a trade-off between data quality and diversity that applies to both the initialization set and the distilled set.
>
> Unlike existing parameterized approaches [8–9], which require carefully designing neural decoder architectures, including the number of layers, feature dimensions, and other architectural choices, and often struggle to match their parameter counts to the IPC constraint, **GSDD only requires specifying the GPC**. This greatly simplifies both storage and computation, and **the optimal GPC value can be estimated reliably**.
>
> > **Q4:** Gaussian Escape: In your ablation (Table 3), the "w/o boundary" setting shows a performance drop. Is this drop due to removing the loss term (Eq 10) or the hard clipping (Eq 11)?
>
> **Summary:** We provide more fine-grained ablation studies on boundary constraints.
>
> In Table 3, "w/o boundary" indicates the removal of **both** the loss term and hard clipping.
>
> We designed these two operations for distinct purposes:
>
> 1. **clip:** Prevents Gaussian points from "escaping" the visual area where they would lose gradient signals and be permanently lost. However, points may still cluster at the boundaries, degrading central feature representation.
> 2. **reg (Regularization):** A soft constraint encouraging points to disperse more toward the image center.
>
> Table 9: Detailed Ablation on Boundary Constraints
>
> |              | ACC  |
> | ------------ | ---- |
> | w/o reg      | 76.2 |
> | w/o clip     | 76.2 |
> | w/o reg+clip | 75.1 |
> | w/ reg+clip  | 76.4 |
>
> The results show that both the regularization loss and hard clipping contribute independently and positively to the final distillation performance.
>
> Most of the results have already been updated in the manuscript, and in particular, our ablation study on GPC is presented with a more intuitive visualization in Figure 6.
>
> We once again thank you for your recognition and valuable suggestions. We are happy to answer any further questions.
>
> References:
>
> [1] Dataset Condensation via Efficient Synthetic-Data Parameterization, ICML, 2022
>
> [2] Frequency Domain-Based Dataset Distillation, NeurIPS, 2023
>
> [3] Generalized Large-Scale Data Condensation via Various Backbone and Statistical Matching, CVPR, 2024
>
> [4] Elucidating the design space of dataset condensation, NeuRIPS, 2024
>
> [5] Distilling dataset into neural field, ICLR. 2025
>
> [6] Dataset distillation with neural characteristic function: a minmax perspective, CVPR, 2025
>
> [7] Towards trustworthy dataset distillation: a benchmark of privacy, fairness and robustness, IJCNN, 2024
>
> [8] Dataset Distillation via Factorization, NeurIPS, 2022
>
> [9] Sparse Parameterization for Epitomic Dataset Distillation, NeurIPS, 2023
>
> [10] Dataset distillation by matching training trajectories, CVPR, 2022
>
> [11] Dataset Condensation With Distribution Matching, WACV, 2023
>
> [12] Squeeze, Recover and Relabel: Dataset Condensation at ImageNet Scale From A New Perspective, NeurIPS, 2023

---

### Author Response · Authors · 2025-12-02
**Summary of Rebuttal**

This paper introduces Gaussian-based representations into the field of dataset distillation (GSDD), and its novelty, efficiency, and strong performance have been unanimously recognized by the reviewers.

- The reviewers mainly raised the following **core concerns** regarding GSDD:
  - **Lack of large-scale dataset distillation.** To address this concern, we have supplemented experiments at 256 resolution on the six ImageNet subsets (in Table 3), as well as experiments on ImageNet-1K at 224 resolution (in Table 18). To the best of our knowledge, this is the first work to introduce a distilled dataset parameterization method to ImageNet-1K.
  - **Lack of in-depth analysis of GS parameterization.** To address this concern, we benchmarked different parameterization methods in terms of representativeness and efficiency, and GSDD achieves the best trade-off (in Table 5). Leveraging both advantages, GSDD can boost performance by representing more distilled images (increasing Gaussian Images Per Class) under the same storage (fixed Images Per Class) without compromising representation quality or exceeding memory constraints (in Figure 5 and 6).
- For most of the **remaining concerns** raised by the reviewers, we provide detailed experiments and explanations, including: initialization (in Table 17), cross-platform robustness of the rasterization cuda operator (in Table 20), extensive ablations on hyperparameters and modules (in Figure 6, Table 6, 13 and 14), cross-resolution distillation experiments (in Table 21) and so on.
- Several interesting ideas were also raised, such as exploring cross-image Gaussian representations and extending the framework to other modalities. We have provided our internal experimental insights and detailed explanations, and GSDD could be expanded along these directions in future work.

Due to the new policy, we have not yet communicated with the reviewers. However, we have already conducted detailed experiments and explanations. Below are our responses to each reviewer’s questions:

---

> ### Author Response · Authors · 2025-12-02
> **Summary of Rebuttal (Part 1/4): Reviewer  sSwA (Initial Rating: 4, Confidence: 4)**
>
> Reviewer sSwA praised GSDD as novel, efficient and high-performing, but raised main concerns regarding the lack of large-scale experiments, confounding initialization, hyperparameter complexity.
>
> > **Concern 1 (Large-scale experiments):** Require experiments on ImageNet-1K.
>
> * **Solution:** We reported results on ImageNet-1K ($224\times 224$) in **Table 18** and six subsets ($256\times 256$) in **Table 3**. We also implemented IDC as a basic baseline on ImageNet-1K.
> * **Conclusion:** This paper first introduces a distilled dataset parameterization method to ImageNet-1K. We show SOTA performance on ImageNet-1K and six subsets.
>
> > **Concern 2 (Confounding Initialization):** Require experiments without initializing from real images.
>
> * **Solution:** We provided results without initialization in **Table 17**, still outperforming most methods with initialization.
> * **Conclusion:** We clarified that the initialization strategy is a common technique in dataset distillation. And our method could still show strong performance without it.
>
> > **Concern 3 (Hyperparameter Complexity):** Require further ablation study on GPC.
>
> * **Solution:** We provided a detailed ablation study on GPC with different IPC, datasets, and distillation baseline in **Figure 6**. We showed the initial performance after initialization from real images shows strong correlation with final performance.
> * **Conclusion:** Optimal GPC could be estimated from the intial performance, and performance around optimal GPC is robust. GSDD is more hyperparameter-friendly than other methods which need additional network structure information.
>
> > **Other concerns:** We provided more fine-grained results on the boundary-constraint ablation study in **Table 14**.

---

> ### Author Response · Authors · 2025-12-02
> **Summary of Rebuttal (Part 2/4): Reviewer  45iu (Initial Rating: 4, Confidence: 5)**
>
> Reviewer 45iu praised GSDD as novel, promising, and high-performing, but raised main concerns regarding the lack of analysis on why GSDD uses Gaussian Splatting, why GSDD achieves high performance in depth, and whether GSDD could be applied across diverse application scenarios.
>
> > **Concern 1 (Lack of motivation in depth):** Require experiments to demonstrate advantages of GSDD compared to prior methods.
>
> * **Solution:** We benchmarked representation performance  against prior methods on image reconstruction tasks in **Table 5**.
> * **Conclusion:** GSDD shows both strong efficiency and representation capacity compared to prior methods.
>
> > **Concern 2 (Insufficient evidence for strong performance):** Require experiments to analyze the reason behind high performance.
>
> * **Solution:** We provided results with varying GPC in **Figure 6**, and it turned out GPC greatly contributes to performance.
> * **Conclusion:** Based on the low GPU memory utilization and fast optimization, we could scale GPC up and achieve remarkable performance, which is hard for other methods due to the trade-off between efficiency and representation capacity.
>
> > **Concern 3 (No analysis on other applications):** Require extending GSDD to other application scenarios such as cross-resolution, corrupted datasets, other modalities.
>
> * **Solution:** We provided results on cross-resolution in **Table 21**, which first obtains a distilled dataset in low resolution and then evaluate on high resolution.
> * **Conclusion:** Despite performing worse than DDiF, GSDD shows less performance degradation than other methods by supersampling. GSDD has potential to extend to other modalities by using the GS framework, which is left for future work.
>
> > **Other Concerns:** We provided more detailed ablation study on modules in **Table 6, 13**, performance under high resolution ($256\times 256$) in **Table 3**, performance with soft-label, performance without bf16, and potential improvements.

---

> ### Author Response · Authors · 2025-12-02
> **Summary of Rebuttal (Part 3/4): Reviewer  7jwx (Initial Rating: 4, Confidence: 4)**
>
> Reviewer 7jwx praised GSDD for being well written, well motivated, simple, effective, high-performing and universal, but raised main concerns regarding cross-platform robustness of the rasterization operator, cross-architecture performance, and scalability of GSDD.
>
> > **Concern 1 (Cross-platform robustness of the rasterization operator):** Require empirical evidence across different GPU architectures.
>
> * **Solution:** We benchmarked the rasterization operator on 5 GPU architectures including 3090, 4090, 5090, V100 and A100 in **Table 20**.
> * **Conclusion:** Results show consistent GPU memory ($\pm 0.00$) and performance ($\pm0.41$), except a slight performance drop on V100.
>
> > **Concern 2 (Cross-architecture performance):** Require more detailed cross-architecture results.
>
> * **Solution:** We added per-architecture comparison in **Table 16**.
> * **Conclusion**: Although GSDD does not outperform the SOTA method in the fine-grained comparison, it achieves state-of-the-art performance in the overall evaluation.
>
> > **Concern 3 (Scalability of GSDD):** Require results on large-scale experiments.
>
> * **Solution:** We reported results on ImageNet-1K ($224\times 224$) in **Table 3** and six subsets ($256\times 256$) in **Table 18**. We also implemented IDC as a basic baseline on ImageNet-1K.
> * **Conclusion:** This paper first introduces a distilled dataset parameterization method to ImageNet-1K. We showed SOTA performance on ImageNet-1K and subsets dataset.

---

> ### Author Response · Authors · 2025-12-02
> **Summary of Rebuttal (Part 4/4): Reviewer  zWtU (Initial Rating: 4, Confidence: 3)**
>
> Reviewer zWtU praised GSDD as the first work to introduce Gaussian framework into the DD field, and the contribution of the customized CUDA operators, but raised main concerns regarding large-scale experiments, intrinsic advantages of 2D Gaussians.
>
> > **Concern 1 (Large-scale experiments):** Require experiments on high resolution datasets.
>
> * **Solution:** We validated GSDD on six ImageNet subsets ($256\times256$) in **Table 3**.
> * **Conclusion:** We showed SOTA performance on six ImageNet subsets ($256\times256$).
>
> > **Concern 2 (Intrinsic advantages over other methods):** Need formal analysis of the advantages of GSDD compared to prior methods.
>
> * **Solution:** We benchmarked representation performance with prior methods on image reconstruction tasks in **Table 5**.
> * **Conclusion:** GSDD shows both strong efficiency and representation capacity compared to prior methods.
>
> > **Concern 3 (Formalization of superiority of GSDD):** Similar to concern 2 and concern 1&2 of reviewer 45iu.
>
> > **Concern 4 (Ablation studies on key hyperparameters):** Require an ablation study on GPC and initial strategy.
>
> * **Solution:** We provided a detailed ablation study on GPC with different IPC, datasets, and distillation baseline in **Figure 6**. We showed that the initial performance after initialization from real images has strong correlation with final performance. We provided performance with random initialization and warm-up initialization in **Table 17**.
> * **Conclusion:** Optimal GPC could be estimated from the initial performance, and performance around optimal GPC is robust. GSDD is more hyperparameter-friendly than other methods. We clarified initialization strategy is a common technique in dataset distillation. Our methods still show strong performance without initializing from real images.
>
> > **Concern 5 (Cross-architecture):** Require cross-architecture results.
>
> * **Solution:** We added explicit references to **Appendix B.2** in the main text.
>
> > **Other concerns:** We have fixed typos in the paper.

---

> ### Author Response · Authors · 2025-12-02
> **Appreciation for the Reviewers' Valuable Feedback**
>
> We have uploaded a revised PDF that includes the detailed setup, results, and explanations for the new supplementary experiments, with all updates highlighted in red. We hope that GSDD can inspire further research and contribute to the dataset distillation community.
>
> We sincerely thank all reviewers for their constructive feedback and for recognizing our study. We would also like to invite the reviewers to look over the responses and revised manuscript. We hope these revisions provide a more comprehensive evaluation and a deeper understanding of GSDD.

---

### Meta-Review · Area_Chair_GYYg · 2026-01-01

**Summary:**

This paper introduces Gaussian-based representations into the field of dataset distillation (GSDD). The main concerns are lack of clear justification for the motivation, nsufficient evidence to support the claim, limited analysis. A rebuttal is provided to partially address these concerns, but a major revision is needed for publication in top conferences.

**Reviewer Concerns:**

Concerns of Reviewer sSwA is partially addressed.
Concerns of Reviewer 45iu is not addressed.
Concerns of Reviewer 7jwx is partially addressed.
Concerns of Reviewer zWtU is not addressed.

**Reviewer Scores:**

Reviewer sSwA may change their score to 6.
Reviewer 45iu would not change their score.
Reviewer 7jwx would not change their score.
Reviewer zWtU would not change their score.

---

### Decision · Program_Chairs · 2026-01-26

Reject